# Recombinant Rift Valley fever viruses encoding bluetongue virus (BTV) antigens: Immunity and efficacy studies upon a BTV-4 challenge

Sandra Moreno[1], Eva Calvo-Pinilla[1], Stephanie Devignot[2], Friedemann Weber[2], Javier Ortego[1]*, Alejandro Brun[1]*

**1** Centro de Investigación en Sanidad Animal (CISA), Instituto Nacional de Investigación y Tecnología Agraria y Alimentaria (INIA), Valdeolmos (Madrid), Spain, **2** Institute for Virology, FB10-Veterinary Medicine, Justus-Liebig University, Giessen, Germany

* ortego@inia.es (JO); brun@inia.es (AB)

**Data Availability Statement:** All relevant data are within the manuscript and its Supporting information files.

## Abstract

### Background

Many ruminant diseases of viral aetiology can be effectively prevented using appropriate vaccination measures. For diseases such as Rift Valley fever (RVF) the long inter-epizootic periods make routine vaccination programs unfeasible. Coupling RVF prophylaxis with seasonal vaccination programmes by means of multivalent vaccine platforms would help to reduce the risk of new RVF outbreaks.

### Methodology/Principal findings

In this work we generated recombinant attenuated Rift Valley fever viruses (RVFVs) encoding in place of the virulence factor NSs either the VP2 capsid protein or a truncated form of the non-structural NS1 protein of bluetongue virus serotype 4 (BTV-4). The recombinant viruses were able to carry and express the heterologous BTV genes upon consecutive passages in cell cultures. In murine models, a single immunization was sufficient to protect mice upon RVFV challenge and to elicit a specific immune response against BTV-4 antigens that was fully protective after a BTV-4 boost. In sheep, a natural host for RVFV and BTV, both vaccines proved immunogenic although conferred only partial protection after a virulent BTV-4 reassortant Morocco strain challenge.

### Conclusions/Significance

Though additional optimization will be needed to improve the efficacy data against BTV in sheep, our findings warrant further developments of attenuated RVFV as a dual vaccine platform carrying heterologous immune relevant antigens for ruminant diseases in RVF risk areas.

**Funding:** AB is supported by grants AGL2014-57430R and AGL2017-83226R from the Spanish Ministry of Science, and grant S2018/BAA 4370 from Comunidad de Madrid R&D in Technologies Program. JO is supported by grants AGL2014-57430R and AGL2017-82570R from the Spanish Ministry of Science and Innovation and European Union's Horizon 2020 research and innovation program under grant proposal 727393-2. SMF is a recipient of a pre-doctoral fellowship program from the Spanish Ministry of Science and innovation. FW is supported by the CCHF Vaccine consortium that has received funding from the European Union's Horizon 2020 research and innovation program under grant agreement no. 732732, and by the Swedish Research Council (Vetenskapsrådet, VR) section Research Environment, Infection Biology (contract number 2018-05766). The funding sources had no involvement in the study design nor in the writing of the report and the decision to submit the article for publication.

**Competing interests:** The authors have declared that no competing interests exist.

## Author summary

Live attenuated Rift Valley fever (RVF) vaccines constitute a reliable intervention measure to reduce the burden of the disease in endemic countries. In this work we report the generation of attenuated Rift Valley fever virus (RVFV) that express vaccine antigens of bluetongue virus (BTV) instead of the virulence factor NSs. The recombinant viruses were able to induce protective immune responses against both RVFV and BTV when administered as vaccines in mice and sheep respectively. Though further optimization is needed to enhance the level of protection in sheep upon a single dose, these results demonstrate the potential of attenuated RVFV as a vaccine vector for other ruminant diseases, in this case enabling bluetongue vaccination while immunizing against RVF. Since RVF outbreaks are sporadic events, preventive vaccination is often not perceived as a real need. In such scenario a bivalent vaccine strategy would make RVF vaccination more appealing.

## Introduction

Rift Valley fever virus (RVFV) is a mosquito-borne pathogen causing severe disease outbreaks affecting humans and livestock in sub-Saharan Africa, Egypt, Yemen, Saudi Arabia and the Indian Ocean islands [1]. In Africa, the re-emergence of Rift Valley fever (RVF) is usually associated to heavy rainfall seasons favoring overgrowth of mosquito species capable to transmit the virus trans-ovarially. It has been proposed that the virus is able to resist desiccation for long periods within mosquito eggs, which hatch upon flooding starting a new infectious cycle [2,3]. RVFV belongs to the Phlebovirus genus from the *Phenuiviridae* family, Order Bunyavirales. It is an enveloped, single-stranded RNA virus with a segmented genome of negative and ambisense polarity [4]. The virus particles contain three genomic RNA segments: the L (large) segment encodes the viral RNA-dependent RNA polymerase (RdRp); the M (medium) segment codes for a polyprotein precursor that is further processed into two surface glycoproteins (Gn and Gc), a 13-14kDa non-structural protein (NSm) that has been shown to play a role in the suppression of apoptosis in infected cells [5], and a 78kDa polypeptide (comprising NSm and Gn sequences) that is incorporated into virions produced in infected insect cells [6]; the S (small) segment encodes in an ambisense orientation the viral nucleoprotein N and the non-structural protein NSs, considered a virulence factor responsible of host general transcription suppression, the suppression of antiviral interferon (IFN)-β gene activation and the facilitation of viral translation [7,8]. The NSs protein is not essential for virus replication. In fact, natural NSs deletion mutants were found with an attenuated or avirulent phenotype [9] due to the presence of an activated IFN system [7,8]. One of these NSs deletion mutants, Clone 13, constitutes a live attenuated vaccine derivative considered now as a promising control measure in African countries. In spite of the availability of effective RVF vaccines in Africa, the sporadic, often unpredictable, RVF outbreaks after long inter-epizootic periods makes annual vaccination programs difficult to be established. A potential strategy to overcome this problem would be the use of multivalent vaccines that could provide immunity against a prevalent disease for which vaccination is mandatory while immunizing against other diseases of more sporadic nature.

Bluetongue (BT) is a viral hemorrhagic disease of ruminants, causing high morbidity and mortality, highly prevalent in tropical and subtropical regions coinciding with the presence of competent *Culicoides* vectors [10]. Bluetongue virus (BTV) outbreaks have been reported in several European, Asian and American countries and is considered an endemic disease in

temperate settings with established vector populations. BTV is an Orbivirus that belongs to the *Reoviridae* family. The BTV genome consists of ten linear dsRNA segments that encode seven structural and five non-structural (NS) proteins [11–13]. The outer VP2 capsid protein is the main target for virus neutralizing antibodies [14] whereas the non-structural NS1 protein contains epitopes associated with both cellular and humoral responses [15]. Effective control of BT relies in vaccination campaigns against the prevalent or seasonal BTV serotype using available modified-live and inactivated vaccines, although these types of vaccines still offer opportunities for improvement [16].

The use of RVF live attenuated vaccines in the field is limited in endemic and non-endemic countries, due to common concerns related to incomplete attenuation and/or potential reassortment events. Reverse genetics allowed to rescue RVFV entirely from cDNA [17–19] and RVF viruses with modified genomes lacking virulence associated, non-structural virus genes constitute now relevant vaccine candidates [20–27]. The possibility of manipulating the RVFV genome by means of reverse genetic systems brought also an opportunity for generating attenuated RVFV that encode and express heterologous genes [28]. In this work, using our reverse genetics system [19], we generated recombinant RVFV able to express either a truncated BTV-4 NS1 protein (amino acids 1 to 270) or the VP2 capsid antigen. The recombinant viruses were characterized phenotypically and tested for their ability to induce protective immune responses against both RVFV and BTV antigens in mice and sheep as well. Our data strongly supports the feasibility of using RVFV as a viral vaccine vector for ruminants; in this example enabling vaccination against a prevalent bluetongue disease while immunizing against Rift Valley fever.

## Methods

### Ethics statement

All experiments with live animals were performed under EU guidelines (Directive 86/609) upon approval by the Ethical Review and Animal Care Committees of INIA and Comunidad de Madrid (permit code PROEX 037/15).

### Viruses and cells

The cell lines used for this study were HEK293T (human embryonic kidney 293 cells, ATCC CRL-3216), Vero (African green monkey kidney cells, ATCC CCL-81) and BHK-21 (baby hamster kidney fibroblasts, ATCC CCL-10). All cell lines were grown in Dulbecco's modified Eagle medium (Invitrogen) supplemented with 10% fetal bovine serum (FBS), 2 mM L-glutamine, 1% 100X non-essential amino acids, 100 U/ml penicillin, 100 µg/ml streptomycin. All cells were incubated at 37° C in the presence of 5% $CO_2$. BTV serotype 4 Morocco strain MOR2009/09 (BTV-4M) [29], BTV serotype 4 SPA2004/02 (BTV-4), RVFV strain 56/74 and RVFV-MP-12 viruses were used in the experiments.

### Mice

Type-I interferon receptor defective mice (IFNAR[-/-]) on a 129 Sv/Ev background (A129), wild-type 129 Sv/Ev and BALB/C were used for the in vivo studies. Mice were housed in biosafety level 3 (BSL-3) animal facilities at INIA-CISA before use.

### Plasmid construction

Plasmids pI.18_RVFV_L, pI.18_RVFV_N, pHH21_RVFV_vL, pHH21_RVFV_vM, pHH21_RVFV_vN_GFP and pHH21-RVFV-vN_TCS were previously described [19].

Plasmids pHH21-RVFV-vN-VP2, pHH21-RVFV-vN-NS1, and pHH21-RVFV-vN-NS1NtV5tag were derived from pHH21-RVFV-vN_TCS. The heterologous sequences were inserted into the TCS (tandem cloning site) region via two *AarI* restriction sites that generate NcoI/XhoI-compatible ends (Fig 1A).

The sequences of the BTV-4 heterologous genes were amplified by PCR from pcDNA3-VP2 using primer pairs (S1 Table) VP2_NcoI_fwd and VP2_XhoI_rev or from pSC11-NS1 using primer pairs NS1_NcoI_fwd and NS1_XhoI_rev (for the generation of pHH21-RVFV-vN-NS1) and NS1_NcoI_fwd and NS1Nt_XhoI_rev (for the generation of pHH21-RVFV-vN-NS1Nt) (Fig 1B). The NS1 gene has a *XhoI* site that had to be removed in order to be inserted into the TCS. This *XhoI* site was removed from pSC11-NS1 by PCR using primer pairs NS1_fw_GxC and NS1_rev_GxC. The generation of the pHH21-RVFV-vN-NS1NtV5tag was carried out using the pSC11-NS1 that had been already modified to remove the *XhoI* site, using primer pairs NS1Nt_V5tag_fwd and NS1Nt_V5tag_rev to insert the V5tag epitope at the C-terminus of NS1Nt.

## Rescue of recombinant rRVFVs expressing heterologous BTV antigens

To recover recombinant RVFV with the pol I/II rescue system [19], HEK293 and BHK-21 cells were seeded in a 1:1 ratio in six-well plates. Semi-confluent layers of co-cultured cells were transfected with 1 μg each of pHH21-RVFV-vL, pHH21-RVFV-vM, pHH21-RVFV-vN_GFP

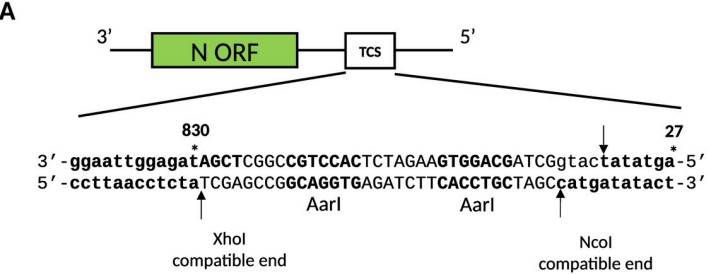

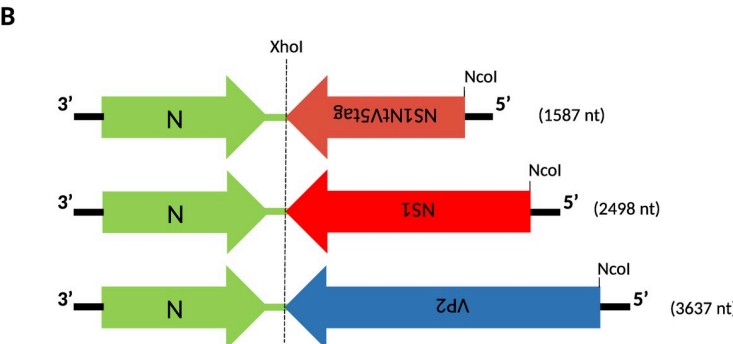

**Fig 1. Outline for the generation of recombinant RVF viruses. A**. Schematic representation of the cloning site of plasmid pHH21-RVFV-vN_TCS based on [19]. The viral N gene and the tandem cloning site (TCS) are symbolised by boxes. Nucleotide sequence of the TCS is shown. The two *AarI* sites were used for the insertion of BTV-4 ORFs. Digestion with *AarI* restriction enzyme generates compatible N*co*I and *XhoI* recognition sequences, indicated by lines and arrows. Nucleotide positions (*) refer to the full-length S virus segment in positive-sense. **B**. Schematics of plasmids showing different inserted BTV-4 sequences. Inverted writing denotes the coding orientation with respect to the nucleoprotein ORF in the viral genome. Nucleotide numbers indicates the size of the inserted foreign sequences.

pHH21-RVFV-vN-VP2 pHH21-RVFV-vN-NS1Nt, (which provide the viral segments in negative-sense (i.e. genomic polarity) under control of a human pol I promoter) and with 0.5 μg of pI.18-RVFV-L and pI.18-RVFV-N (to provide the viral N and L proteins for packaging and replication of the viral genomic RNAs) using 13.5 μL of Lipofectamine 3000 (Invitrogen). The supernatants of transfected cells were harvested on day 3, 5 and 7 post-transfection. To screen for the presence of recombinant virus, Vero cells grown in six-well plates were inoculated with 500 μL of each supernatants. Cytopathic effect (CPE) detection, GFP expression or immuno-fluorescence with specific antibodies at 4 days post-infection (dpi) indicated the presence of recombinant virus infection. As a positive control, the parental rZH548 virus was also rescued. Rescued viruses were further propagated in cell cultures. At post-transfection passage 2, the presence of rescued virus was confirmed by plaque assay on Vero cell cultures and the virus encoding VP2 transgene was plaque purified twice. Further two passages were performed to generate a virus stock for use in the present experiments.

## Plaque assay

Vero cells grown in 12-well plates were inoculated with ten-fold serial dilutions of supernatants from infected cells in DMEM with 10% FBS. After 1 h incubation at 37˚C, the inoculum was removed and cells were overlaid with semi-solid medium (Eagle's Minimun Essential Medium (EMEM)-1% carboxymethylcellulose) and further incubated for 72–96 h at 37˚C with rZH548ΔNSs::GFP, rZH548ΔNSs::VP2$_{BTV-4}$ or 5 days at 37˚C for rZH548ΔNSs:: NS1Nt$_{BTV-4}$. Cells were fixed with 10% formaldehyde, stained with 2% crystal violet and titres were calculated from the plaque numbers obtained for each dilution.

## In vitro expression analysis of heterologous genes

In order to verify the expression of the heterologous genes, Vero cells were infected with each recombinant virus. At 48 hours post-infection cells were fixed with 1:1 methanol-acetone solution and incubated 20 minutes at -20˚C. Methanol-acetone was removed and PBS-FBS 20% added and incubated for one hour at room temperature. After washing once with PBS the primary antibody was added (diluted 1:200 in PBS-FCS 20% for serum and 1:500 for monoclonal antibody). A polyclonal mouse serum was used for the detection of VP2 [30], a mouse monoclonal antibody against the V5 epitope tag (clone MCA 1360, Bio-Rad) was used for the detection of NS1Nt and a polyclonal rabbit anti-RVFV serum [31] was used for the detection of RVFV proteins. Primary antibodies were incubated for one hour at 37˚C or 4˚C overnight. After incubation with the primary antibody, cells were washed three times with PBS-T (PBS with 0.05% Tween-20) and incubated for 30 min at room temperature with goat anti-mouse Alexa Fluor 488 or goat anti-rabbit Alexa Fluor 594 (Thermo), followed again by three PBS-T washes.

For western blot analysis, monolayers of Vero cells were infected with rZH548-ΔNSs:: VP2$_{BTV-4}$ or rZH548-ΔNSs::NS1Nt $_{BTV-4}$. After 24 hours, cells were washed twice with PBS and lysed in SDS-PAGE sample buffer. Samples were analysed by SDS-PAGE in reducing conditions. Proteins were transferred to nitrocellulose membrane (Whatmann). The membrane was incubated for one hour with 5% skimmed milk in Tris-buffered saline (TBS). Upon blocking, anti-V5 epitope mAb or the 2B1 anti-RVFV-N mAb were added diluted 1:1000 in 5% milk-TBS-T (TBS with 0.05% Tween-20) and incubated at 4˚C overnight. The membranes were then washed three times with TBS-T and incubated for 1 hour at room temperature with an anti-mouse IgG-HRPO conjugated antibody. The membrane was washed again three times with TBS-T before adding luminescent substrate (ECL-GE Healthcare). Gel images were visualized using a ChemiDoc Imaging System (Bio-Rad).

For immunoprecipitation analysis monolayers of Vero cells were infected with rZH548-ΔNSs::VP2$_{BTV-4}$, rZH548-ΔNSs::NS1Nt $_{BTV-4}$, BTV-4 or MP12 virus. At 24 hpi the inoculum was removed and 200μCi of [S35]-Methionine-Cysteine solution (Thermo) in Met-Cys free medium were added to each well. 24 hours later the cells were lysed in radio-immunoprecipitation assay (RIPA) buffer and frozen at -80˚C until use. Protein-G beads (Invitrogen) were incubated with rotation with 15 μl of polyclonal serum from BTV convalescent mice immunized with MVA expressing NS1/VP2/VP7 [32] in a final volume of 200 μl of PBS for 1 hour at room temperature. The bead-Ab immunocomplexes were washed once with PBS-T and then 250μl of the radiolabelled lysate were added and incubated with rotation overnight at 4˚C. Upon three additional washes, Laemmli's buffer was added and the samples boiled for 5 minutes. The immunocomplexes were then separated by 12% SDS-PAGE. Fixed gels were subjected to fluorographic enhancement using Amplify solution (Amersham, GE). Dried gels were exposed to X-ray films for several hours and developed.

## RT-qPCR assays

Mouse blood samples were collected at specific times upon challenge and RNA was extracted using a RNA virus extraction kit (Speedtools, 180 Biotools BM). Real-time RT-qPCR specific for BTV segment 5 and/or RVFV L-segment were performed as described [33,34]. To establish a correspondence between Cq values and plaque forming units (pfus), blood from naïve mice were spiked with $10^1$ to $10^5$ pfus of either BTV-4 or RVFV and RNA was extracted for RT-qPCR as above.

## Analysis of genetic stability

Vero cell supernatants upon a first, second, third and fourth serial passages of each recombinant virus were titrated and used to infect Vero cell monolayers using a MOI of 1. 72 hours later supernatants were collected and RNA extracted using a RNA virus extraction kit as above. Superscript IV (Thermo) was used for reverse transcription of RNA from rZH-548-ΔNSs::VP2$_{BTV-4}$ infection using a genomic S-segment specific primer. The resulting cDNA was amplified by PCR using primer pairs for the nucleoprotein N gene and the heterologous VP2 gene (S1 Table). RNA extracted from ZH548-ΔNSs::NS1Nt$_{BTV-4}$ infected cell culture supernatants was used to perform one-step real time-PCR with primer pairs specific for the NS1 gene and the L segment of RVFV as described elsewhere [15,35]. In this case the amplification products were detected using Taqman 5' FAM probes (S1 Table).

## ELISA

ELISA plates were coated overnight at 4˚C with the antigen of interest (either BTV-4 infected cell lysates or purified recombinant RVFV-N protein) diluted in carbonate/bicarbonate buffer, pH 9.6. After blocking at 37˚C with 5% skim milk in PBS, the primary antibody was added, diluted in 5% skim milk in PBS and incubated for 1 hour at 37˚C. After three consecutive washes with PBS, the secondary antibody, a HRPO-conjugated goat-anti-mouse, was added and incubated for one hour at 37˚C. After incubation and washing steps, the TMB substrate (Sigma) was added for 10 min, followed by one volume of stopping solution (1N H$_2$SO$_4$). Optical densities were measured at 450 nm.

## Assessment of humoral immune responses against rZH548-ΔNSs::VP2$_{BTV-4}$

Groups of BALB/c mice (n = 4) were inoculated intraperitoneally with $10^7$, $10^4$ or 10 pfu of rZH548-ΔNSs::VP2$_{BTV-4}$. Blood samples were collected at days 7 and 14 post-inoculation to

examine the induction of neutralizing antibodies. Four groups were inoculated with $10^5$ pfu of BTV-4 at day 14. Blood samples were taken again at days 7 and 14 post-viral boost and were used in sero-neutralization assays. Four groups were then challenged with 500 pfu of virulent RVFV-56/74 strain and mice were maintained under observation for clinical signs and mortality for two weeks.

### RVFV neutralization assay

Sera from vaccinated mice were heat-inactivated at 56˚C for 30 minutes and two-fold diluted (starting in 1:20) in DMEM and mixed with an equal volume (50 µl) of medium containing $10^3$ pfu of a viral stock (RVFV-MP12 strain). After one hour of incubation at 37˚C, this mixture was added to Vero cell monolayers seeded in 96-well plates. After 3 days at 37˚C in the presence of 5% $CO_2$, cells were fixed and stained with a solution containing 2% crystal violet in 10% formaldehyde. The titre of neutralizing antibodies was determined as the highest dilution that reduced the CPE by 50%.

### BTV-4 PRNT$_{50}$ neutralization assay

Serum samples from mice were heat-inactivated for 30 min at 56˚C and ten-fold serial dilutions were incubated with 100 pfu of BTV-4 for 1 h at 37˚C. Vero cells in 12-well plates were infected with the serum and virus mixture and incubated for 1 hour; then the inoculum was removed and fresh DMEM with 0.4% agar was added. After 5 days at 37˚C, 5% $CO_2$ the plates were fixed and stained with 2% crystal violet solution in 10% formaldehyde. The titre of neutralizing antibodies was determined as the highest dilution that reduced the number of plaques by 50%.

### Passive immunisation with donor antiserum

Donor antisera were prepared by pooling individual serum samples from mice inoculated with the recombinant viruses or BTV-4. The donor antisera were inactivated 30 min at 56˚C and used for intraperitoneal injection of groups of four A129 (IFNAR $^{-/-}$) mice. 100 µl of each antiserum was administered 1 hour before subcutaneous challenge with 500 pfu of BTV-4.

Positive control mice were administered with pooled sera from BTV-4 challenged mice from previous experiments at the same time as the recipients. Clinical signs and viremia were evaluated in all the animals following challenge.

### Analysis of cellular immune responses to rZH548-ΔNSs::NS1Nt$_{BTV-4}$

Two groups of 129 Sv/Ev (n = 4) mice were inoculated with $10^7$ pfu of rZH548-ΔNSs:: NS1t$_{BTV-4}$ intraperitoneally. Three weeks later one of the groups and an additional group of naïve mice received a booster dose with $10^5$ pfu of BTV-4. Two weeks after the last immunization, mice were sacrificed and their spleens collected for intracellular cytokine staining assay (ICCS).

Spleen cells (3 x$10^6$) were stimulated with 10 µg/ml of a NS1-specific peptide or control peptide for 6 hours. For assay performance and background controls, cells were also stimulated with ConA (8 µg/ml) or maintained with complete culture medium, respectively. After stimulation, cells were washed, stained with surface markers, fixed, permeabilized and stained intracellularly using appropriate fluorochromes. For immunostaining, the following fluorochrome-conjugated anti-mouse antibodies were used: anti-CD8-PerCP, anti-IFNγPE and anti-CD4-FITC (Miltenyi). Data were acquired by FACS analysis on a FACSCalibur (Becton

Dickinson). Data analyses were performed using the FlowJo software version X0.7 (Tree Star, Ashland, OR). The number of lymphocyte-gated events was $5 \times 10^4$.

For ELISPOT assay, Immobilon-P plates (Millipore MAIPS-4510) were coated with 50μl of anti-mouse IFNγ capture antibody (AG-18/RA-6A2, BD). After overnight incubation the plates were blocked with RPMI complete medium for 1 hour at 37˚C. A total of $5 \times 10^5$ spleen cells were added to each well and re-stimulated with peptides #152 (10 μg/ml) and #14 (10 μg/ml) for 18–20 hours at 37˚C with 5% $CO_2$. As controls, cell were stimulated either with PHA (8 μg/ml) or with RPMI complete medium. After incubation the plates were washed with $ddH_2O$ and 5 times with PBS. A biotin-conjugated anti-mouse IFNγ antibody (R46A2, BD) was then added and incubated for 2 hours. The PBS wash step was repeated and then peroxidase-conjugated streptavidin was added an incubated for 1 hour at RT. After a last PBS wash 3-amino-9-ethylcarbazole substrate (BD Elispot AEC substrate) was added and incubated at RT until spots were visible. Spots were quantified using an AID EliSpot Reader (Autoimmun Diagnostika, GmbH).

## Sheep immunization

Three groups of four sheep (autochthonous Churra breed) of approximately 4–5 months old were inoculated subcutaneously (sc) with $10^7$ pfu of both recombinant viruses or with ZH548-ΔNSs::GFP expressing virus (negative control). Blood and serum samples were collected at 4, 7, 14 and 21 days post-inoculation for the assessment of neutralizing antibodies. At 21 days post-inoculation sheep were challenged sc with $10^6$ pfu of BTV-4M. Clinical signs were monitorized for two weeks by veterinary personnel blinded to the experimental settings. A clinical score was given for the severity of signs (from 0–3) based on explorations of oral and nasal tissues, (including discharges), postural (inflammation of coronary band, lameness) or behavioural signs (depression, weakness). Viremia titres and antibody levels were estimated in blood and sera samples taken at different time points after challenge.

## BTV-4 TCID$_{50}$ neutralization assay

Sera from vaccinated sheep were heat-inactivated at 56˚C for 30 minutes, two-fold diluted (starting in 1:2) in DMEM and mixed with an equal volume (50 μl) of medium containing $10^3$ pfu of BTV-4. After one hour of incubation at 37˚C, trypsinized Vero cells in a volume of 100 μl were added to this mixture in 96-well plates. After 5 days at 37˚C in the presence of 5% CO2, cells were fixed and stained with 2% crystal violet solution in 10% formaldehyde. The titre of neutralizing antibodies was determined as the highest dilution showing 50% of CPE reduction.

## Biochemical determinations in blood samples

Whole blood samples were collected at different time points post BTV-4 challenge in BD Vacutainer tubes for serum clotting. Serum was separated by centrifugation (1500 x g for 10 min) and kept to -20˚C until use. Alanine aminotransferase (ALT), aspartate aminotransferase (AST), lactate dehydrogenase (LDH), gamma-glutamyltransferase (GGT), alkaline phosphatase (ALP), albumin (ALB), bilirubin (BIL), blood urea nitrogen (BUN) and total protein (TP) were measured in sheep sera in a Saturno 100 analyser (Crony Instruments, Rome, Italy) using specific reagents according to the manufacturer's instructions (Spin React, Vall D'En Bas, Spain). The activity of enzymes was expressed as international units per litre (IU/l), serum proteins in g/dL and organic compounds in mg/dL.

## Statistical analysis

Statistical analyses were performed using GraphPad Prism version 6.0 (GraphPad Software, San Diego, CA). Differences in antibody levels, T-cell responses, serum biochemical parameters, fever and viremia between vaccine groups were calculated by ANOVA tests. Survival data were analyzed using a log rank test with mice grouped by immunization strategy. A significance level of $P < 0.05$ was set in all analyses.

## Results

### Rescue and characterization of recombinant Rift Valley fever viruses expressing BTV proteins

Our RNA pol I/II promoter-driven reverse genetic system [19] was used to rescue recombinant RVFV. The genes encoding either the full-length BTV-4 VP2 or NS1 proteins were amplified from plasmids encoding BTV-4 VP2 or NS1 genes using forward and reverse primers including NcoI and XhoI restriction sites at 5' respectively (S1 Table). The PCR products were purified and inserted into AarI digested-plasmid pHH21-RVFV-vN_TCS in which the NSs sequence was replaced with a unique tandem cloning site (TCS) (Fig 1A). Recombinant NSs deleted viruses (RVFV-ΔNSs) were generated in co-cultured BHK-21 and HEK293 cells by co-transfection with mixtures of five plasmids (as described in the Methods section) containing either VP2 or NS1 genes. (Fig 1B).

As a control of virus rescue experiments, recombinant RVFV-ZH548 (rZH548wt) and RVFV-ZH548-ΔNSs::GFP (rZH548-ΔNSs::GFP) were rescued upon plasmid transfection. Extensive cytopathic effect (CPE) was observed after serial blind passages of cell culture supernatants from the transfection mixture that included the plasmid encoding BTV-4 VP2 gene, indicative of the potential rescue of a recombinant rZH548-ΔNSs::VP2$_{BTV-4}$ virus. In contrast, all attempts made to rescue virus from mixtures that included the plasmid encoding BTV-4 NS1 gene were unsuccessful since no evident CPE was detected upon any subsequent passages. To overcome potential toxic effects induced by full-length NS1 gene expression, a truncated version of the NS1 gene encoding the N-terminal half of the protein (amino acids 1 to 270) in frame with the V5 epitope tag (IPNPLLGLD) at the C- terminus (NS1Nt-V5) was also cloned into plasmid pHH21-RVFV-vN_TCS. The use of this transfection mixture produced an evident CPE after serial cell culture passages, indicating the putative rescue of a rZH548-ΔNSs::NS1Nt$_{BTV-4}$ virus. The presence of recombinant virus in the supernatants was further confirmed by plaque forming assay in Vero cells (as described in Methods section). To check whether the rescued viruses were expressing the heterologous genes, infected cultures were analyzed by immunofluorescence using either a polyclonal anti-BTV-4 VP2 serum raised in mice or a monoclonal antibody anti-V5 tag (Fig 2A). VP2 expression was predominantly cytoplasmic (Fig 2A, upper panel) while NS1Nt (Fig 2A, bottom) expression was observed in the form of discrete granular or aggregated bodies. Western blot analysis of infected cell lysates showed that rZH548-ΔNSsNS1Nt $_{BTV-4}$ expressed NS1Nt in agreement with the predicted size of 31kDa (Fig 2B). In contrast, the expression of intact VP2 was barely detected (Fig 2C, arrowheads). Instead, a major 25kDa form was observed upon immunoprecipitation of rZH548-ΔNSs::VP2$_{BTV-4}$ infected Vero cells (Fig 2C). The presence of VP2 viral mRNA was analyzed by RT-PCR from rZH548-ΔNSs::VP2$_{BTV-4}$ infected cell cultures (S1A and S1B Fig). PCR fragments were later subjected to Sanger sequencing to detect the presence of potential frameshifts that could explain the synthesis of the 25kDa form. However, no changes affecting the full VP2 ORF were found by this analysis.

The *in vitro* growth kinetics for the recombinant viruses were compared with that of recombinant wt RVFV (rZH548) and the GFP-expressing NSs deletion mutant rZH548-ΔNSs::GFP,

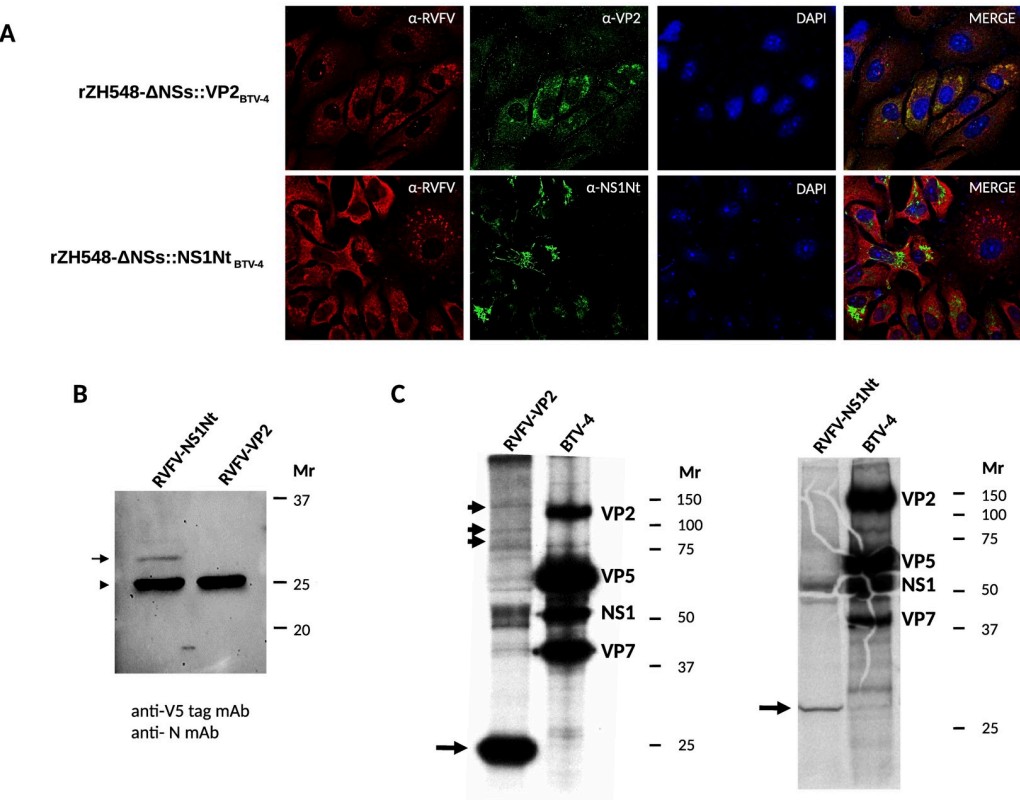

**Fig 2. Detection of recombinant BTV-4 gene expression in RVFV infected cell cultures. A**. Indirect immunofluorescence analysis (IFA) of rescued RVFVs infected Vero cells. A rabbit anti-RVFV polyclonal serum was used for the detection of RVFV antigens (red fluorescence). Mouse anti-VP2 polyclonal serum was used for VP2 detection and an anti-V5 mAb was used for the detection of NS1Nt (green fluorescence). **B**. Western blot analysis of rZH548-ΔNSs:: NS1Nt $_{BTV-4}$ infected Vero cells (lane 1) or rZH548-ΔNSs::VP2 $_{BTV-4}$ (lane 2). Anti-V5tag mAb was used for the detection of NS1Nt (arrow). As a control of infection, nucleoprotein N (arrowhead) was detected using 2B1 mAb. Mr: relative molecular mass in kDa. **C**. Immunoprecipitation analysis of rZH548-ΔNSs::VP2$_{BTV-4}$ (RVFV-VP2), rZH548-ΔNSs:: NS1Nt$_{BTV-4}$ (RVFV-NS1Nt) or BTV-4 infected cell lysates (BTV-4). Mouse anti-BTV polyclonal serum was used for the detection of BTV specific antigens. Images were overexposed to visualize upper putative VP2 protein bands (arrowheads). Major immunoprecipitated products in both rRVFV lysates are indicated by arrows. The identity of proteins immunoprecipitated in the control BTV-4 lysate is indicated. Mr: relative molecular mass in kDa.

in Vero and HEK293 cells using low MOI (0.01 PFU/cell). Supernatants were harvested at different times post-infection and virus titers were determined by plaque assay in Vero cells (Fig 3). Independent of the cell line used, growth rates and end titers of both rRVFVs expressing BTV-4 proteins were similar to those of rZH548-ΔNSs::GFP, but were consistently lower if compared with the recombinant wt rZH548 virus (Fig 3A). All three ΔNSs viruses evidenced turbid plaques, slower kinetics and lower titers in HEK293 cells, consistent with the IFN-competent phenotype of these cells [19]. Interestingly, rZH548-ΔNSs::VP2$_{BTV-4}$ and rZH548-ΔNSs::NS1Nt $_{BTV-4}$ viruses showed different plaque phenotypes when compared to rZH548-ΔNSs::GFP. (Fig 3B).

## Analysis of the stability upon cell passage of rZH548-ΔNSs::VP2$_{BTV-4}$ and rZH548-ΔNSs::NS1Nt $_{BTV-4}$

Since the foreign sequences introduced in the genome of the recombinant viruses altered the normal size and gene sequence of the small (S) RNA genomic segment, we tested whether this

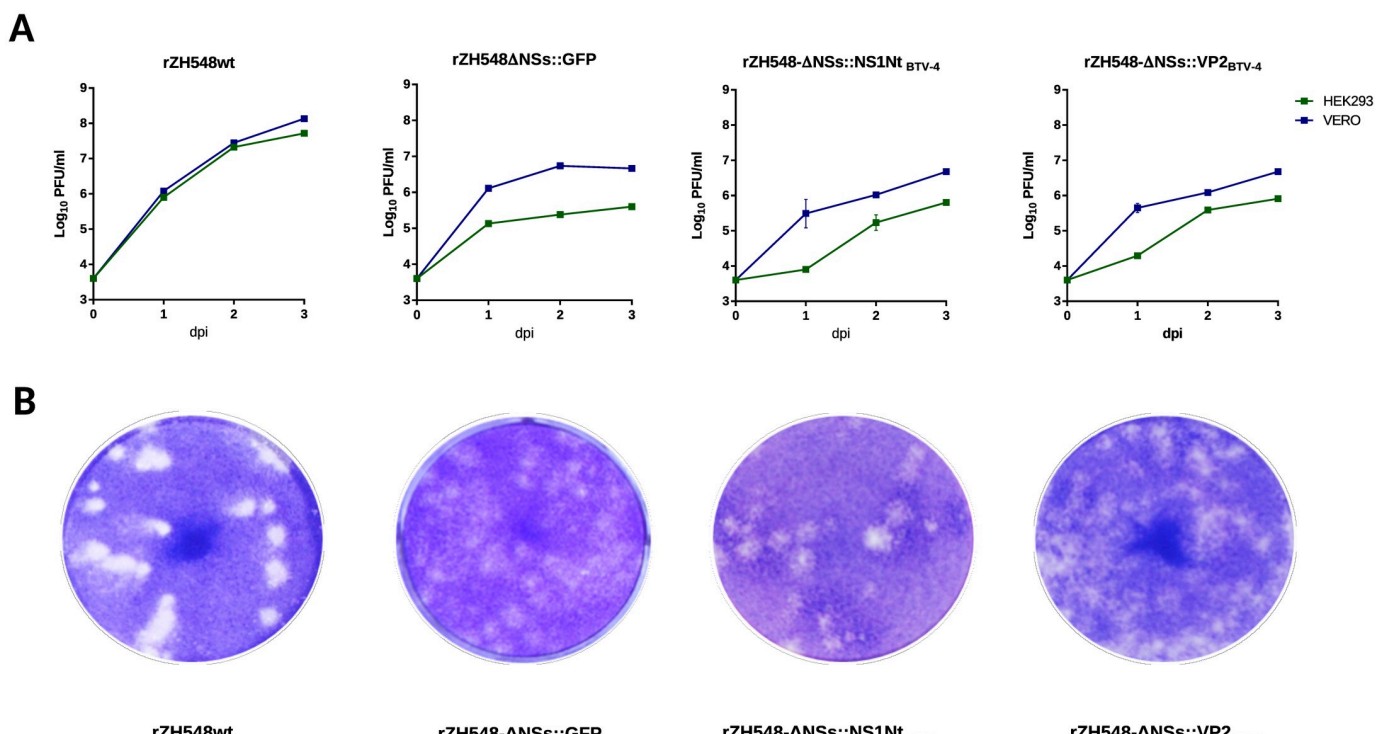

**Fig 3. Phenotypic characterization of the recombinant viruses. A**. Growth kinetics of rRVFVs. Vero or HEK293 cells were infected with a MOI of 0.01 and supernatants were harvested at the indicated time points. Viral titers were determined by infection of Vero cells with serial dilutions of the supernatants. **B**. Plaque phenotypes of the rescued recombinant viruses. Cell lysis was visualized at 3–5 days post infection upon Crystal Violet staining.

novel genomic architecture remained conserved upon four serial passages in cell culture. Using RNA extracted from infected cell culture supernatants with different passage history we analyzed the presence of both S-segment transgenes by either a quantitative (NS1 gene) or gel-based (VP2 gene) PCR (S2A Fig) The detection levels for BTV-NS1 or BTV-VP2 gene sequences were normalized to those of RVFV viral RNA polymerase (RdRp) gene (L-segment) or the nucleoprotein N gene (S segment), respectively, to provide relative detection values. After four consecutive passages both transgenes were detected with consistent ratios respect to the RdRp or nucleoprotein genes (S2B and S2C Fig). Most importantly, CPE was observed along each passage together with the detection of the transgene expression by indirect immunofluorescence using specific antibodies (S2D Fig). These data were consistent with genomic stability of both viruses in terms of transgene integrity along a number of cell passages.

## Assessment of infectivity and immunogenicity of recombinant rZH548-ΔNSs::VP2$_{BTV-4}$ and rZH548-ΔNSs::NS1Nt $_{BTV-4}$ in mice

RVF virus lacking the NSs gene are highly attenuated in type I IFN-competent mice, and induce a strong immune response shortly after inoculation [9,36]. To confirm that both rescued rZH548-ΔNSs::VP2$_{BTV-4}$ and rZH548-ΔNSs::NS1Nt $_{BTV-4}$ maintained a non-virulent phenotype, BALB/c mice were inoculated intraperitoneally with a high dose ($10^7$ pfu) of either rZH548-ΔNSs::VP2$_{BTV-4}$ or rZH548-ΔNSs::NS1Nt $_{BTV-4}$. All mice survived the challenge without apparent clinical signs inducing similar titers of anti-N antibodies as judged by ELISA (S3 Fig), indicating that the attenuated phenotype was not affected by the encoded VP2 or NS1Nt

transgenes. The induction of anti-N humoral responses was dose dependent, increasing significantly along time as shown by the kinetics of anti-N IgGs upon infection with rZH548-ΔNSs::VP2$_{BTV-4}$ (Fig 4A). To test whether the inoculation induced protection against a virulent RVFV challenge, these mice were subsequently inoculated with a lethal dose of the RVFV-56/74 virulent strain three weeks after the first inoculation with rZH548-ΔNSs::VP2$_{BTV-4}$. All mice survived with no clinical signs, except those mice belonging to the mock inoculated (PBS) control group, which died between 4–6 days (Fig 4B). At day 3 post RVFV-56/74 challenge high viral RNA loads were detected in the mice from the control group and lower levels in the group inoculated with 10 pfu of the recombinant virus, while the mice from the other groups showed no detectable viraemia (Fig 4C).

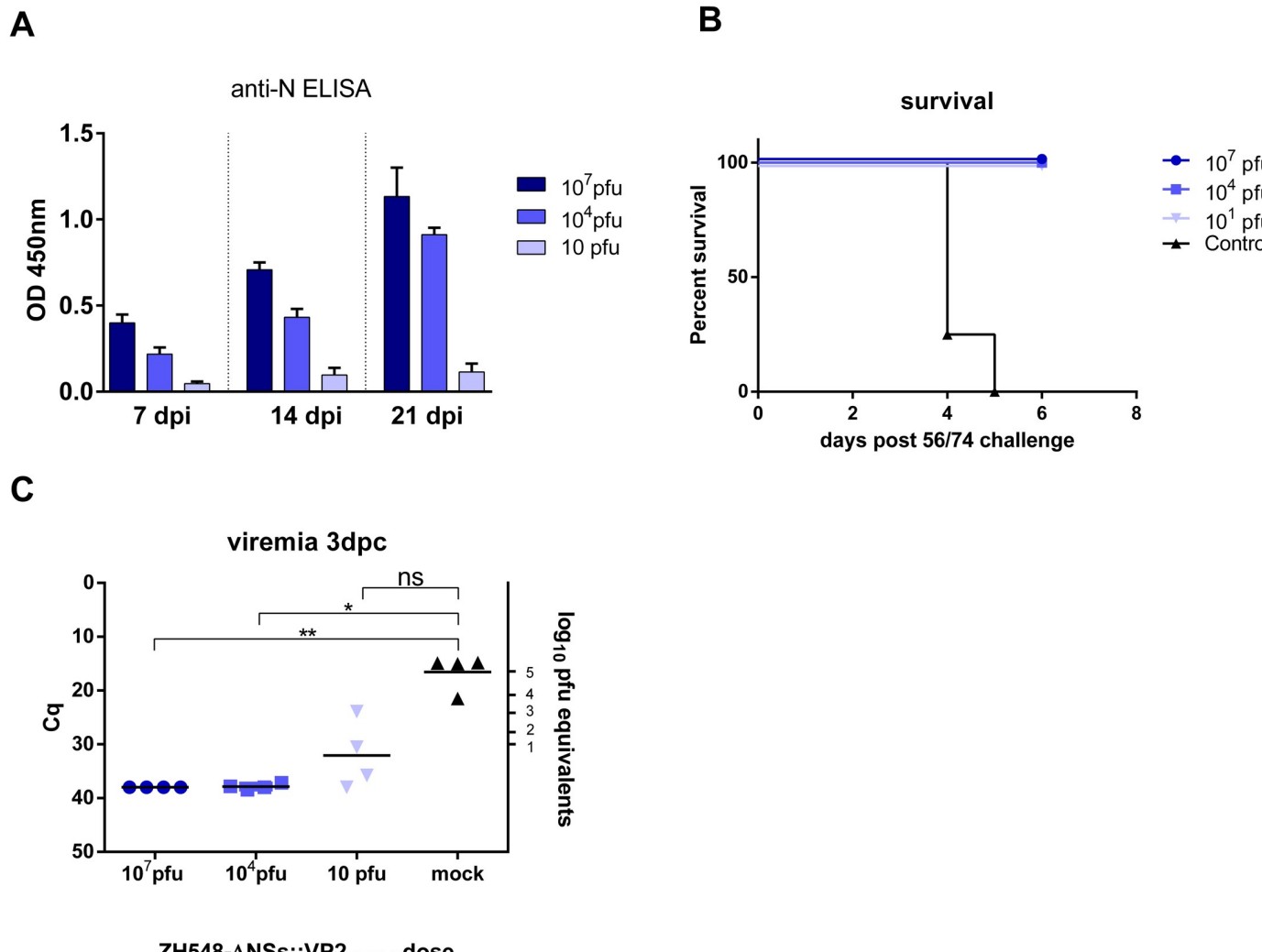

**Fig 4. *In vivo* infectivity of the recombinant rZH548-ΔNSs::VP2$_{BTV-4}$ virus. A**. Kinetics of anti-RVFV nucleoprotein antibodies in serum from mice inoculated with different doses of rZH548-ΔNSs::VP2$_{BTV-4}$. Bars represent median plus interquartile range. **B**. Survival curves of mice (n = 4) after intraperitoneal inoculation at the indicated doses of rZH548-ΔNSs::VP2$_{BTV-4}$. **C**. Viremia titers by RT-PCR at day 3 after RVFV 56/74 challenge. The correlation of Cq (quantification cycle) data with pfu equivalents is indicated in the right Y axis. The data are plotted for individual mice (n = 4) with means (horizontal lines). *p<0.05; **p<0.01; ***p<0.001 (Anova, non-parametric Kruskall-Wallis test).

## Analysis of anti-BTV protective antibody responses upon inoculation of mice with rZH548-ΔNSsVP2$_{BTV-4}$

To analyze whether the inoculation of rZH548-ΔNSs::VP2$_{BTV-4}$ induced anti-VP2 antibodies, ELISA plates were coated with a BTV-4 infected cell lysate and the presence of serum anti-VP2 antibodies in mice determined. The mice inoculated with the highest dose of the virus ($10^7$ pfu) showed significantly increased amounts of anti-VP2 antibodies with respect to the groups receiving lower doses ($10^4$ and $10^1$ pfu) indicating a dose-dependent antibody induction (Fig 5A). To confirm the induction of VP2-specific antibodies, the sera were used in a BTV-4 sero-neutralization assay. Here, titers of BTV-4 neutralizing antibodies (nAb) were observed in the sera of mice vaccinated with $10^7$ and $10^4$ pfu. However, nAb titers peaked at 14 days post inoculation (dpi) dropping by 21dpi in the sera from both groups. In contrast, no neutralizing antibodies were detected in animals immunized with only 10 pfu, suggesting that the induction and duration of serum nAbs was also dose dependent (Fig 5B). To test if the first dose of rZH548-ΔNSs::VP2$_{BTV-4}$ administered to the mice was able to prime adequately for further memory responses after a second antigen exposure, a dose of $10^5$ pfu of BTV-4 virus was inoculated 21 days after the initial inoculation. The booster dose with BTV-4 increased the titers of anti-VP2 antibodies in the $10^7$ pfu group as judged by ELISA (Fig 5A). In contrast, all groups increased the levels of nAbs as measured 7 days after boosting. Interestingly, the group of mice immunized with the lowest dose (10 pfu) showed higher levels of neutralizing antibodies than the control, indicative of an effective immune priming by the recombinant virus (Fig 5B).

BALB/C mice elicited humoral anti-VP2 responses upon rZH548-ΔNSs::VP2$_{BTV-4}$ inoculation as shown by ELISA or seroneutralization assays. Since anti-VP2 nAbs associate with protection against BTV [14] the inoculation of this recombinant virus could have conferred protection against a BTV-4 challenge to the mice. However, since BALB/C mice are refractory to BTV-4 infection, we had no means for a direct estimation of the efficacy of this vaccine strategy against a BTV-4 challenge. To overcome this obstacle we performed a passive serum transfer experiment in BTV-sensitive A129 mice (IFNAR$^{-/-}$) using pooled sera from groups of BALB/C mice immunized with (i) $10^7$ pfu of rZH548-ΔNSs::VP2$_{BTV-4}$ (prime only group); (ii) $10^7$ pfu of rZH548-ΔNSs::VP2$_{BTV-4}$ and $10^5$ pfu of BTV-4 (prime-boost group); (iii) $10^5$ pfu of BTV-4 (control group); and (iv) mock immunized mice (negative control). After passive transfer, each mouse was challenged with a lethal BTV-4 dose of $5x10^2$ pfu. Only the mice that received serum from the prime-boost approach survived the challenge, without showing any clinical manifestation. In contrast, the mice receiving serum from either BTV-4 or prime only mice succumbed to the challenge, although with a slight delay with respect to the control mice (Fig 6A). At day three post-challenge all groups showed lower viremia levels than the mock inoculated group and those of BTV or prime-boost groups reached statistical significance (Fig 6B). In agreement with the survival data the lowest viral RNA loads were found in those mice from the prime boost group. This result suggest that, as for BTV-4, a single dose of rZH548-ΔNSs::VP2$_{BTV-4}$ induces humoral immunity to levels that reduce viremia without conferring a fully protective response. However rZH548-ΔNSs::VP2$_{BTV-4}$ works efficiently as a priming agent as shown by the efficacy after a booster dose.

## Analysis of the cellular immune response induced upon inoculation of rZH548-ΔNSs::NS1Nt $_{BTV-4}$ in mice

The BTV-4 NS1 protein has been proposed as a candidate antigen to develop multi-serotype BTV vaccines due to its high degree of conservation among most BTV serotypes as well as to the presence of molecular determinants of cellular immunity [15]. Thus, the cellular anti-BTV immune response elicited by the recombinant virus rZH548-ΔNSs::NS1Nt $_{BTV-4}$ was assessed

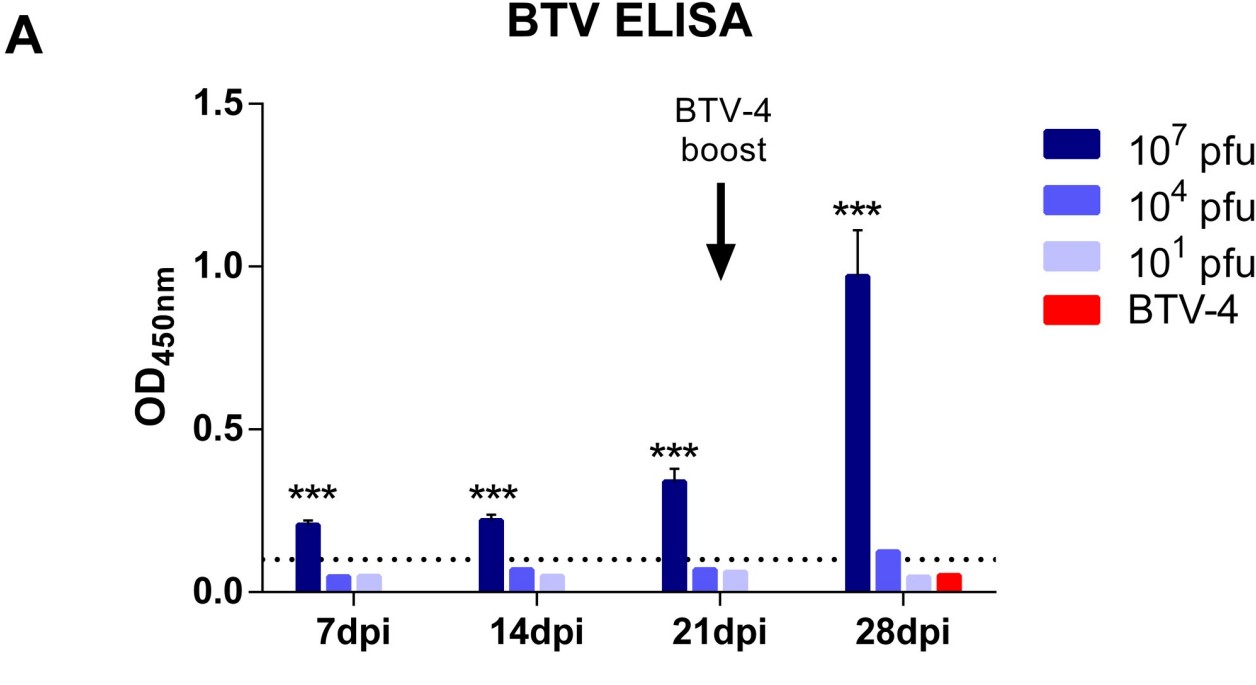

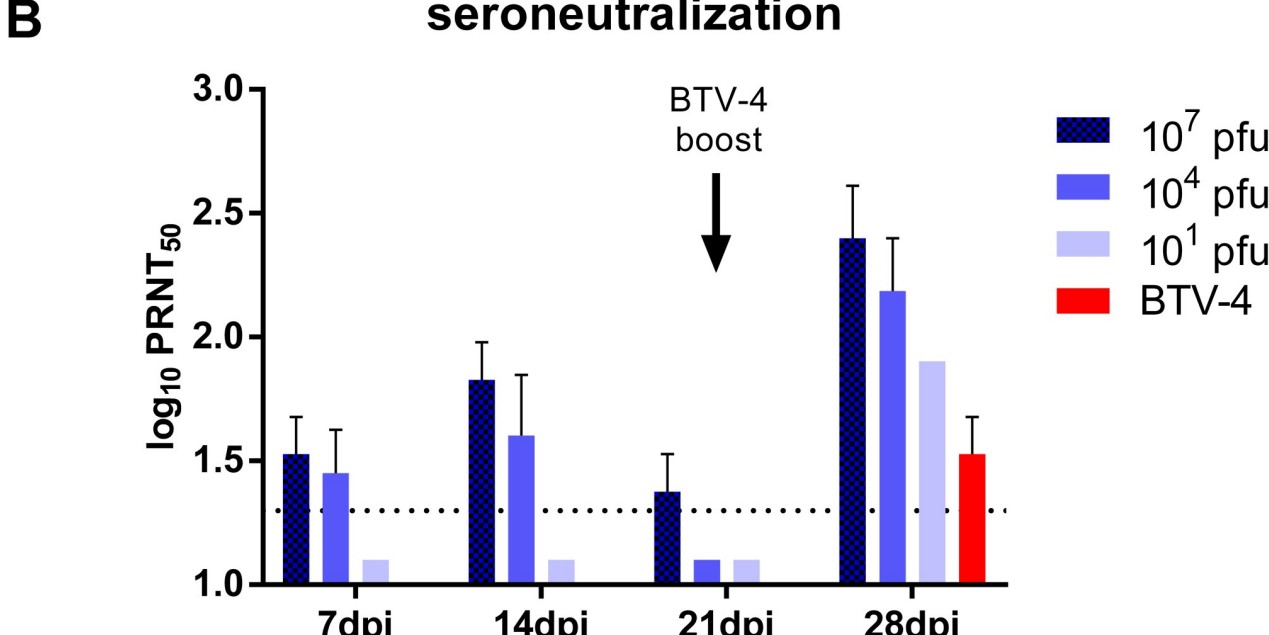

**Fig 5. Anti-VP2 responses upon single dose rZH548-ΔNSs::VP2$_{BTV-4}$ inoculation of mice. A**. Detection by ELISA of anti-BTV serum antibodies upon inoculation of different doses of the recombinant virus. ELISA plates were coated with BTV-4 infected antigen lysate. Serum samples diluted 1/100 were obtained at the indicated days post inoculation and 7 days after a BTV-4 boost (arrow) ***P<0.001 (two-way Anova test). **B**. Kinetics of neutralizing antibody induction before and after boosting with BTV-4 virus (arrow). Horizontal dotted lines represent assays' sensitivity limits.

in mice two weeks after the last immunization in the presence or absence of a booster dose of BTV-4. The mice were sacrificed and spleen cells were isolated to perform intracellular IFN-γ staining (ICCS) of T-lymphocytes and to determine the number of IFN-γ-producing spleen

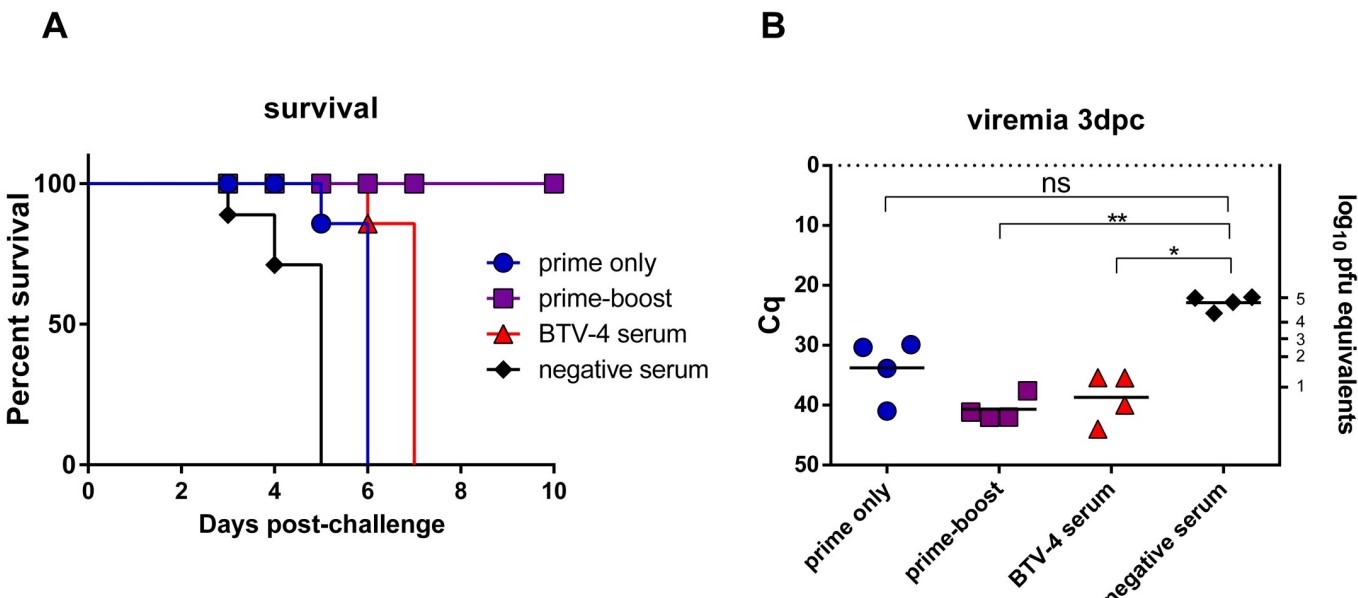

**Fig 6. Passive transfer experiment of immune sera in A129 mice.** Groups of n = 5 mice were inoculated ip with serum pools from BALB/C vaccinated mice with rZH548-ΔNSs::VP2$_{BTV-4}$, before (prime only) and after BTV-4 boost (prime-boost), with anti -BTV-4 mouse serum or with negative mouse serum. **A.** Survival of A129 mice after challenge with BTV-4. **B**. Detection of BTV-4 viremia in A129 mice by rt-PCR at day three post BTV-4 challenge. **P<0.01; P*<0.05 (one way non-parametric Anova). Horizontal dotted line represent the cut-off Cq value. The correlation of Cq (quantification cycle) data with pfu is indicated in the right Y axis.

cells by ELISPOT. Spleen cells were stimulated with the immunodominant CD8+ T cell NS1 peptide (GQIVNPTFI, namely #152), or an irrelevant peptide (#14) to determine the assay background. A significant percentage of IFN-γ+ CD8+ T cells upon stimulation with the peptide #152 was observed in mice primed with rZH548-ΔNSs::NS1Nt $_{BTV-4}$ and boosted with BTV-4 (4,095%), while lower stimulation levels were observed in those mice that were only primed with one dose of rZH548-ΔNSs::NS1Nt $_{BTV-4}$ or BTV-4 (0,7975% and 0,5575%, respectively) (Fig 7A). ELISPOT data confirmed that higher numbers of IFN-γ secreting cells responded to the stimulation with peptide #152 in those mice that were immunized with rZH548-ΔNSs::NS1Nt $_{BTV-4}$ and boosted with BTV-4 compared to the other groups. Interestingly, the rZH548-ΔNSs::NS1Nt $_{BTV-4}$ primed-only group showed higher numbers of stimulated cells with respect to the groups primed with BTV-4 alone although these differences did not reach statistical significance (Fig 7B).

## Immunogenicity and efficacy against BTV challenge of the recombinant viruses in sheep

The data on the attenuation, stability upon cell passage and immunity conferred by the rescued viruses to mice prompted us to test the efficacy of both rZH548ΔNSs::NS1Nt $_{BTV-4}$ and rZH548-ΔNSs::VP2$_{BTV-4}$ as potential vaccines for ruminants. With this aim we set a preliminary vaccination experiment using three groups of n = 4 sheep each. Two groups were vaccinated with $10^7$ pfu of either rZH548-ΔNSs::VP2$_{BTV-4}$ or rZH548ΔNSs::NS1Nt $_{BTV-4}$, and the third group received a similar dose of rZH548-ΔNSs::GFP as a negative control. The kinetics of the antibody response after immunization showed that BTV-4 neutralizing antibodies were present in the serum from the rZH548-ΔNSs::VP2$_{BTV-4}$ vaccinated group as early as 4 dpi, reaching maximum levels at 7 dpi, and clearly decreasing by days 14 and 21 post vaccination

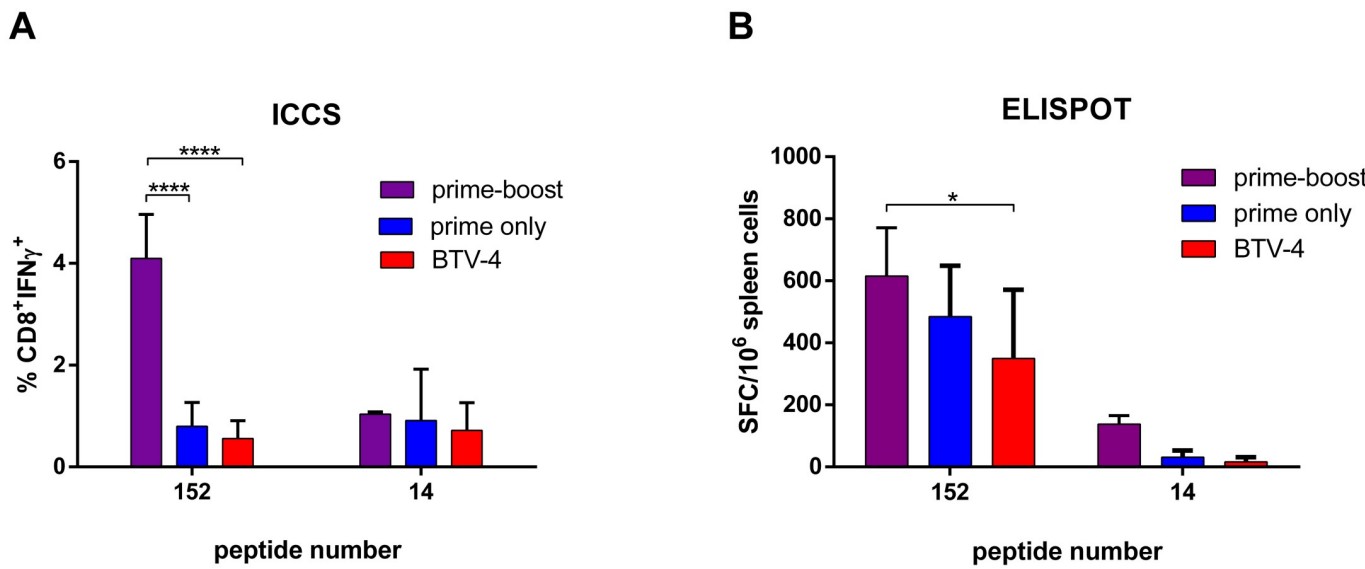

**Fig 7. Analysis of cellular immune responses.** Groups of n = 4 mice were inoculated with BTV-4, rZH548-ΔNSs::NS1Nt$_{BTV-4}$ before (prime only) or after BTV-4 boost (prime-boost) **A.** Intracellular cytokine cell staining (ICCS). Percentage of CD8$^+$ IFN-γ$^+$ cells after re-stimulation with NS1 peptide #152 or peptide #14 (irrelevant peptide). **B.** ELISPOT assay of re-stimulated spleen cells. Bars represent mean plus SD. $^*$P<0.05, $^{****}$P<0.0001 (one way non-parametric Anova).

(Fig 8A). As expected, neutralizing activity was not found in the sera from sheep immunized with rZH548-ΔNSs::NS1Nt $_{BTV-4}$ or rZH548-ΔNSs::GFP. In contrast, RVFV neutralizing antibody titers reached maximum levels by day 14 and were maintained at 21dpi (Fig 8B). The RVFV neutralizing antibody kinetics were similar in all three groups of sheep, irrespective of the vaccine used, indicating that the expression of heterologous antigens was not detrimental for the induction of anti-RVFV humoral immunity in sheep. Interestingly, the level of anti-RVFV-N antibodies detected by ELISA was slightly lower in the sheep vaccinated with rZH548ΔNSs::GFP, although these differences were not statistically significant (Fig 8C).

In order to test the efficacy of the immunity provided, the sheep were challenged with a 10$^6$ pfu dose of the virulent Morocco BTV-4 reassortant strain. Upon virus challenge, vaccination with any of the three recombinant viruses did not abolish the onset of fever although the group of rZH548-ΔNSs::NS1Nt$_{BTV4-}$ immunized sheep showed lower mean temperatures, delayed and reduced onset and duration of pyrexia (Fig 9A). In addition, control animals immunized with rZH548-ΔNSs::GFP showed slightly higher sums of clinical scores than the animals vaccinated with rZH548-ΔNSs::VP2$_{BTV-4}$ or rZH548-ΔNSs::NS1Nt $_{BTV-4}$. (S4 Fig). Kinetics of viremia levels for BTV-4 measured by qRT-PCR indicated Ct values in the rZH548-ΔNSs::GFP group that were significantly lower than those from the other two groups at some time points (Fig 9B). The correspondence between Ct levels and viral titers in blood indicated that rZH548-ΔNSs::GFP reached a viremia equivalent to 10$^4$ pfu/ml while in the other two groups the viremia titers remained below that threshold. Finally, post-challenge sera were tested in a BTV-VP7 ELISA to confirm whether the virus had replicated to different extent between groups. In this ELISA we could observe that anti-VP7 antibody levels where consistently higher in sheep from the rZH548-ΔNSs::GFP group (Fig 9C). Altogether, these data suggest that the BTV-4 replication occurred at lower levels in the animals immunized with the recombinant viruses, especially in sheep vaccinated with rZH548-ΔNSs::NS1Nt $_{BTV-4}$.

Blood chemistry analysis were performed using serum samples collected at different days post vaccination (dpv) and challenge (dpc) (S5 Fig). A decrease in the serum levels of alkaline

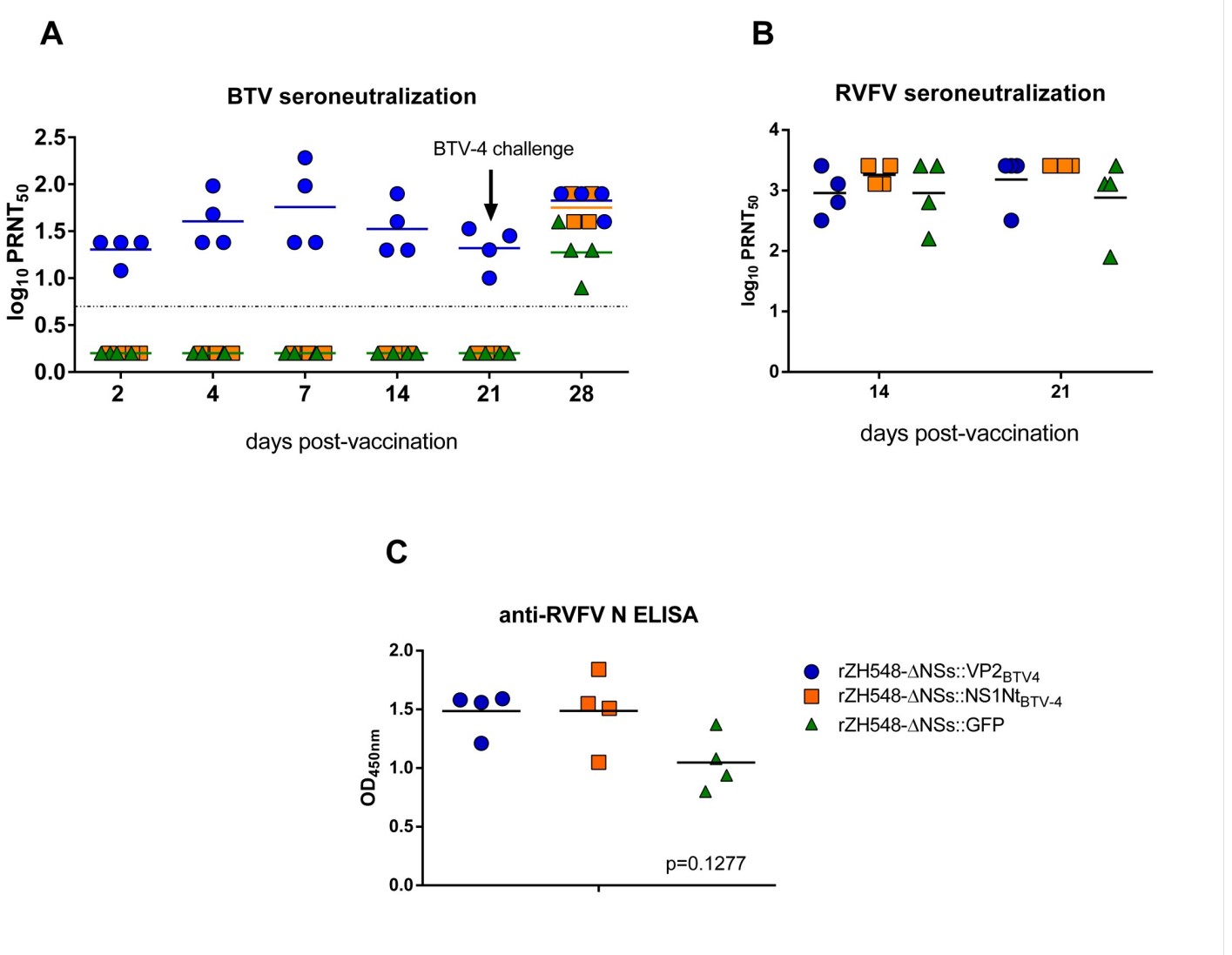

**Fig 8. Antibody responses in sheep upon vaccination with the recombinant viruses. A.** Anti-BTV neutralizing antibody kinetics before (post-immunization) or after challenge with BTV-4 virus measured as 50% infective titers in tissue culture. The limit of sensitivity (horizontal line) was set to 0.69 (1:5 dilution). Values below this threshold were assigned an arbitrary value of 0.2. **B.** Detection of RVFV neutralizing antibodies in sheep sera at 14 and 21 days post-vaccination by plaque reduction neutralization assay ($PRNT_{50}$). The limit of sensitivity (dotted line) was set to 1.3log (1:20 dilution). **C.** Nucleoprotein antibodies in sheep sera at 7 days post-vaccination measured by anti-nucleoprotein ELISA (p value ANOVA test assuming non-parametric distribution). Individual (symbols) and mean values (horizontal bars) are represented. Blue symbols: sheep vaccinated with rZH548-ΔNSs::VP2$_{BTV-4}$ Orange symbols: rZH548-ΔNSs::NS1Nt $_{BTV4.}$ Green symbols rZH548-ΔNSs::GFP.

phosphatase (ALP) and lactate dehydrogenase (LDH) was noted upon vaccination in all sheep. While most of the parameters measured were similar in all groups the BTV challenge had an effect in all sheep by elevating the levels of bilirubin (BIL) and blood urea nitrogen (BUN) a week after, reaching baseline values at day 14 post challenge. Interestingly, animals from both the control and rZH548-ΔNSs::VP2$_{BTV-4}$ groups showed higher mean levels of aspartate aminotransferase (AST) and alanine aminotransferase (ALT) after BTV-4 challenge compared to the animals in rZH548-ΔNSs::NS1Nt$_{BTV-4}$ group, in agreement with the lower impact of the BTV challenge in this group of sheep (S5 Fig).

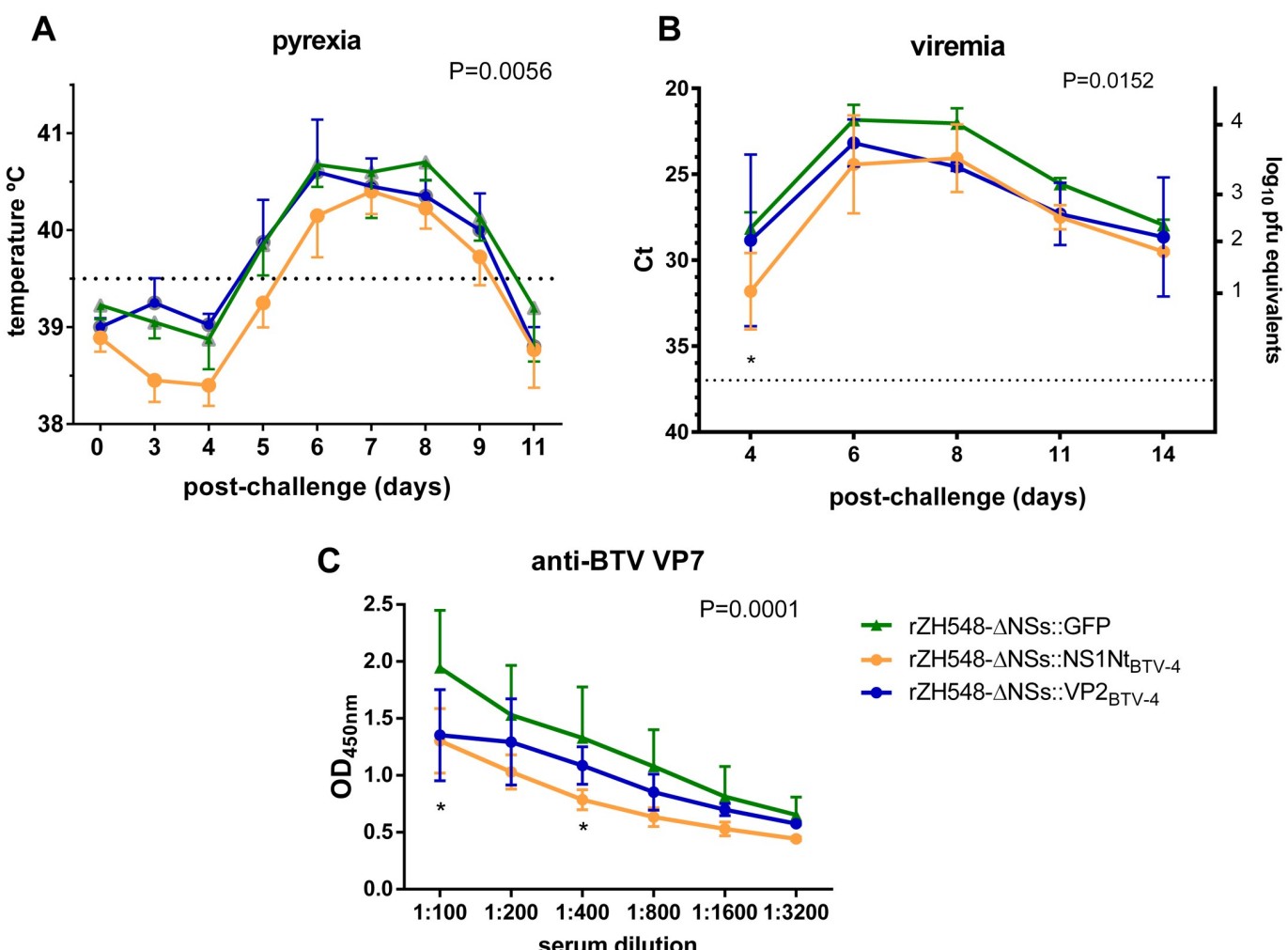

**Fig 9. Efficacy assessment after a virulent BTV-4 challenge in sheep vaccinated with the recombinant RVF viruses. A.** Mean rectal temperatures after BTV-4 challenge. Dotted line represents the fever threshold level determined before challenge. **B.** Viremia analyzed by real time RT-PCR from 4 to 14 dpi. Sheep were monitored daily for two weeks after challenge. The correlation of Ct (threshold cycle) data with pfu equivalents is indicated in the right Y axis. *P<0.05 (multiple comparisons t-test). **C.** Detection of anti- BTV-VP7 antibodies by ELISA in sheep sera taken two weeks after challenge. Mean values (horizontal bars) and standard errors (SEM) are represented. Two-way ANOVA P values (taking vaccine groups as the source of variation) are indicated for each graph. Asterisks denote significant variation (P<0.05) upon Dunett's multiple comparison test.

## Discussion

Vaccination is proposed as a powerful measure to control Rift Valley fever outbreaks in livestock and humans by means of inducing herd immunity [37,38]. The availability of ssRNA(-) virus rescue systems has allowed the manipulation of the RVFV genome [39] and, hence, the generation of several live attenuated experimental RVF vaccines that have proved highly efficacious in ruminants [20–24,26,27]. Previous works using reverse genetics technology also demonstrated the remarkable plasticity of the RVFV genome in accepting foreign insert sequences or even extra genome RNA segments [25,40]. Expression of GFP by means of a recombinant RVFV (rRVFV) was the first demonstration of the viability of a deleted NSs virus expressing a foreign/heterologous antigen [17–19]. Moreover, expression of influenza hemaglutinin (HA) or a tumor-associated CD8 peptide was also possible by means of a non-spreading RVFV replicon [28,41]. Nonetheless, no additional attempts were made to extend the vector potential of

RVFV to other relevant vaccine antigens for ruminants. Our aim since was to explore the feasibility of using RVFV as a vaccine vector for other ruminant viral diseases as well as to provide an appealing strategy to promote vaccination in countries where sporadic RVF disease outbreaks occur. RVF epidemiology is a complex multifactorial process involving susceptible human beings, competent mosquitoes, wild and domesticated animals, and the environment, resulting in severe outbreaks after long inter-epizootic periods with no detectable virus circulation [42,43]. Even if a RVF vaccine was easily available, such unpredictable conditions challenges the implementation of periodic vaccination campaigns, particularly when no policies that can guide utilization of RVF vaccines are provided [44]. A potential approach to overcome this scenario would be to take advantage of periodic vaccination campaigns against more prevalent ruminant diseases, to introduce vaccination against RVF. Bluetongue disease features fit well in this scenario since it is endemic in many countries where RVF epizootics occur and it is far from being eradicated, in particular due to the presence of infected *Culicoides* midges that overcome winter in more temperate regions, favoring recurrent disease outbreaks.

Thus, we first sought to check if RVFV could support the expression of BTV antigens. BTV vaccines rely in the induction of antibody responses against VP2, the major capsid protein and the main inducer of serotype-specific neutralizing antibodies [14]. Other BTV antigen that has shown promise for triggering potential multi-serotype protection was the non-structural NS1 protein. NS1-based protection relies in the elicitation of cross-reactive CTL immune responses [15,45]. NS1 is highly conserved among the various BTV serotypes and it is also the most synthesized viral protein in BTV-infected cells [46]. NS1 contains epitopes associated with both T cell and humoral responses. Antibody responses against NS1 protein may be also important contributors to immune protection even though those antibodies are non-neutralizing. We thus selected both antigens as models to test our hypothesis.

The BTV-4 VP2 antigen was expressed by means of a recombinant RVFV ZH548 virus. The pattern of expression of VP2 in rZH548-ΔNSs::VP2$_{BTV-4}$ infected Vero cells was in agreement with what was seen upon in vitro infection with BTV: an extended cytoplasmic staining with no clear subcellular compartmentalization [47]. Immunoprecipitation analysis of rZH548-ΔNSs::VP2$_{BTV-4}$ in 48hpi infected Vero cells suggested that the VP2 antigen is rapidly degraded to an abundant 25kDa polypeptide, indicative of a poor stability for the native antigen. In this sense, recombinant VP2 expression by means of a modified vaccinia Ankara (MVA) vector showed also a prominent 25kDa degradation product as early as 16 hpi [32]. Moreover, it is known that the outer VP2 protein is highly sensitive to protease digestion rendering, among others, a 25kDa polypeptide [48]. This finding suggested that upon synthesis, VP2 protein is rapidly processed perhaps by specific proteases or, alternatively, due to the lack of other BTV proteins essential to stabilize the VP2 trimer. Nonetheless, the levels of VP2 expression were sufficient to elicit anti-BTV immunity as shown by sero-neutralization data.

On the other hand, expression of the full-length BTV-4 NS1 protein by means of rRVFV infection in Vero cells was unsuccessful, in spite of the number of attempts made, nor it was achieved using a VLP based transcriptional replicon system. It is known that the full-length NS1 protein forms complex tubular-like structures when expressed in BTV infected cells [49] and that NS1 regulates viral gene expression by binding to viral mRNA [50]. In addition, a cytopathogenic role for NS1 has been also suggested [46]. Perhaps the nature of these intracellular structures precluded the RVFV to complete a viral replicative cycle, therefore impeding the generation of infectious progeny particles. Nonetheless, the expression of a truncated version or this protein (NS1Nt) was possible by means of rZH548-ΔNSs::NS1Nt$_{BTV-4}$. It has been shown in previous studies that the majority of the theoretical CD8+ T cell epitopes are located within the amino terminus end of the protein (NS1Nt) [15]. This ensured the presentation of previously reported regions involved in the stimulation of cellular immune responses. The

pattern of expression of NS1Nt was markedly different to that of VP2 and it showed a high resemblance to the pattern observed in BTV infected cells [47] as shown by confocal microscopy data. Vaccination of 129 mice with rZH548-ΔNSs::NS1Nt $_{BTV-4}$ resulted in a NS1 epitope-specific CD8-T cell response in agreement with the immune cell targeting described for RVFV [51].

The use of the RVFV NSs locus to insert foreign genes ensures *a priori* that the resulting rescued virus (lacking NSs expression) should have an attenuated phenotype *in vivo*. We could confirm this attenuation since a high dose (up to $10^7$ pfu/mouse) inoculation of both recombinant viruses was non-virulent in immune competent animals. In contrast, the recombinant viruses retained a highly virulent phenotype in type-I interferon unresponsive A129 mice (IFNAR$^{-/-}$ mice) indicating that their replicative ability *in vivo* was not altered by the expression of the heterologous BTV-4 genes. This fact did not allow us to directly prove the protective efficacy of these recombinant viruses against a BTV challenge since our BTV challenge mouse model relies in the use of IFNAR$^{-/-}$ mice [52,53]. Therefore, we hypothesized that if an appropriate immune response was generated against the heterologous BTV-4 VP2 antigen, the transfer of immune serum could provide a mean to prove that protective immune responses had been elicited. This was supported by the fact that the transferred serum had *in vitro* BTV-4 neutralizing activity. However, transfer of immune serum from rZH548-ΔNSs::VP2$_{BTV-4}$ vaccinated mice was not sufficient to provide full protection (only a slight delay in mortality was observed) indicating that the humoral anti-VP2 immune response was not strong enough or that the amount of serum transferred was still below a protective threshold. However, upon BTV-4 boost the serum was fully protective upon passive transfer. These data pointed out that rZH548-ΔNSs::VP2$_{BTV-4}$ vaccination effectively primed the immune system but perhaps the low expression levels of full-length VP2 limited the induction of stronger anti-BTV humoral immune responses *in vivo*, in spite of the detection of *in vitro* neutralizing antibodies. However, since the transfer of BTV immune sera alone did not provide protection, it could be hypothesized that other mechanisms distinct from humoral immunity need to be stimulated or combined.

We also tested whether our recombinant viruses were able to provide immunity in a large animal model. With this purpose, sheep were inoculated with a dose of $10^7$ pfu and serum antibodies tested for neutralizing activity against both BTV-4 and RVFV. Anti-RVFV neutralizing titers were produced to protective levels as reported elsewhere [54] and, notably, anti BTV-4 neutralizing antibodies were raised faster and to higher titers than those described for other attenuated or inactivated BTV vaccines after two vaccine doses [55,56]. These antibody titers peaked by day seven, decreasing shortly afterwards upon immunization, resembling IgM kinetics. Nonetheless, based on the levels of neutralizing antibody titers induced after a single dose, the rZH548-ΔNSs::VP2$_{BTV-4}$ vaccine appears very efficient providing anti-VP2 immunity. Since the neutralizing antibody titers reached protective levels as reported by others we expected higher degree of protection after the BTV-4 viral challenge in the sheep. The levels detected in the rZH548-ΔNSs::VP2$_{BTV-4}$ group were slightly lower than those of the control group. Sheep vaccinated with rZH548-ΔNSs::NS1Nt $_{BTV-4}$.were better protected against viremia as judged by the levels of anti-BTV VP7 antibodies after challenge. These lower antibody levels could be associated with lower rates of virus replication in those sheep vaccinated with rZH548-ΔNSs::NS1Nt$_{BTV-4}$. Accordingly, ALT and AST levels increased after challenge in both the control and the rZH548-ΔNSs::VP2$_{BTV-4}$ group but not in the rZH548-ΔNSs:: NS1Nt$_{BTV-4}$ group. Though BTV does not target hepatic cells, increased levels of these enzymes might reflect consequences of an inflammatory process triggered upon infection, soft tissue damage and injury to organs such as the lungs [57]. The increased levels observed in this study in sheep upon challenge are in agreement with earlier reports of sheep experimentally

infected with BTV [58]. One explanation for the incomplete efficacy observed in these experiments could be the high challenge dose administered to our sheep, taking in account the particular virulence of the Morocco strain. Another explanation could be that the dose proved to protect mice was as high as the one used for sheep ($10^7$ pfu/animal) so it could be possible that the vaccine dose used in our sheep trial was below an optimal protective threshold. Alternatively, though the nAb titers demonstrated a clear neutralizing activity in vitro, in order to protect sheep from BT it might be necessary to achieve higher affinities upon immunoglobulin subclass-switch for the VP2 specific antibodies. Nonetheless, these data are promising enough to test both recombinant viruses for priming or boosting BTV vaccine doses either alone or in combination. Altogether, they may contribute to support the use of vectored rRVFV-BTV-4 vaccines as a mean to protect against both diseases.

## Supporting information

**S1 Fig. Detection of the VP2 gene by PCR in infected rZH548-ΔNSs::VP2$_{BTV-4}$ or BTV-4 cultures. A**. Schematic of the amplification strategy. Different size VP2 fragments were amplified from cDNA templates. cDNA primers used for first strand synthesis are depicted in red. **B**. Detection of specific fragments by agarose gel electrophoresis.
(TIF)

**S2 Fig. Analysis of the genomic stability of the recombinant viruses. A**. Schematic of the passage history of the recombinant viruses. Vero cell monolayers were infected with of a MOI of 1 with each virus from a first (P1), second (P2), third (P3), or fourth (P4) passage. **B**. Relative detection of L segment RNA (red line) and NS1Nt gene (green) from ZH548-ΔNSs::NS1Nt$_{BTV-4}$ by RT-qPCR (left Y axis). Bars represent the L/NS1 ratio from Ct values (right Y axis). **C**. Relative detection of RVFV nucleoprotein N (red) and BTV VP2 (green) genes from ZH548-ΔNSs::VP2$_{BTV-4}$ by PCR. Bars represent the S-segment/VP2 ratio from densitometry values of gel-based DNA fragments of 521bp and 653bp corresponding to the N and VP2 genes respectively. **D**. Detection of VP2 and NS1 Nt transgenes by immunofluorescence of Vero cells infected with serially passaged recombinant virus (p2 to p5). Mouse anti-VP2 polyclonal serum was used for VP2 detection and an anti-V5 mAb was used for the detection of NS1Nt (green fluorescence). Nuclei were visualized by DAPI staining.
(TIF)

**S3 Fig. Antibody ELISA.** Detection of anti-RVFV nucleoprotein N antibodies 14 days upon inoculation with $10^7$ pfu of rZH548-ΔNSs::VP2$_{BTV-4}$ and rZH548-ΔNSs::NS1Nt$_{BTV4}$ or PBS (mock inoculated control) by ELISA. Bars represent median plus interquartile range. *P<0.05 (Anova, non-parametric Kruskall-Wallis test).
(TIF)

**S4 Fig. Clinical evaluation of sheep.** Clinical display of vaccinated sheep groups at different days after challenge. Scoring was based on the severity of signs observed by blinded veterinary personnel (see Methods). The graph represent the mean (horizontal lines) and min to max values (bars).
(TIF)

**S5 Fig. Blood chemistry analysis in RVFV vaccinated and challenged sheep.** The following biochemical parameters were measured in serum taken at the indicated days post vaccination (dpv) or post challenge (dpc): AST: Aspartate aminotransferase. ALT: Alanine aminotransferase. LDH: Lactate dehydrogenase. GGT: Gamma-glutamyltransferase. ALP: Alkaline phosphatase. ALB: serum albumin. BIL: serum bilirubin. BUN: serum urea nitrogen. TP: serum total

protein. Error bars represent SD. Shadowed areas denote normal sheep enzyme values as described previously [59].
(TIF)

**S1 Table. List of primers used for plasmid construction and RT-PCR analysis.**
(DOCX)

**S1 Data. Excel spreadsheet containing, in separate sheets, the underlying numerical for Figure panels 3a, 4a, 4b, 4c, 5a, 5b, 6a, 6b, 7a, 7b, 8a, 8b, 8c, 9a, 9b, 9c, S2b, S2c, S3, S4 and S5.**
(XLSX)

## Acknowledgments

We thank the animal care and biosafety staff for their excellent technical assistance.

## Author Contributions

**Conceptualization:** Javier Ortego, Alejandro Brun.

**Data curation:** Sandra Moreno, Eva Calvo-Pinilla.

**Formal analysis:** Sandra Moreno, Alejandro Brun.

**Funding acquisition:** Javier Ortego, Alejandro Brun.

**Investigation:** Sandra Moreno, Eva Calvo-Pinilla, Stephanie Devignot, Alejandro Brun.

**Methodology:** Sandra Moreno, Eva Calvo-Pinilla, Stephanie Devignot, Friedemann Weber.

**Project administration:** Javier Ortego, Alejandro Brun.

**Supervision:** Friedemann Weber, Javier Ortego, Alejandro Brun.

**Visualization:** Sandra Moreno.

**Writing – original draft:** Sandra Moreno, Alejandro Brun.

**Writing – review & editing:** Friedemann Weber, Javier Ortego, Alejandro Brun.

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
