## [Decision Letter · Decision Letter 0]

13 Jul 2020

Dear Dr Brun,

Thank you very much for submitting your manuscript "Chimeric Rift Valley fever viruses encoding bluetongue virus (BTV) antigens can provide immunity and partial protection against a BTV-4 challenge in sheep" for consideration at PLOS Neglected Tropical Diseases. As with all papers reviewed by the journal, your manuscript was reviewed by members of the editorial board and by several independent reviewers. In light of the reviews (below this email), we would like to invite the resubmission of a significantly-revised version that takes into account the reviewers' comments. 

Three experts in the field have reviewed your submission. Two of the reviewers had considerable concern with aspect of the presented data and all three felt that significant changes to your manuscript should be made prior to further consideration. 

We would also like to apologize for the extended review time for your submission. The current public health situation has markedly increased demands of our reviewers time.

We cannot make any decision about publication until we have seen the revised manuscript and your response to the reviewers' comments. Your revised manuscript is also likely to be sent to reviewers for further evaluation.

Sincerely,

Michael R Holbrook, PhD

Associate Editor

Amy Gilbert

Deputy Editor

Three experts in the field have reviewed your submission. Two of the reviewers had considerable concern with aspect of the presented data and all three felt that significant changes to your manuscript should be made prior to further consideration. 

We would also like to apologize for the extended review time for your submission. The current public health situation has markedly increased demands of our reviewers time.

Reviewer's Responses to Questions

**Key Review Criteria Required for Acceptance?**

**Methods**

-Are the objectives of the study clearly articulated with a clear testable hypothesis stated?

-Is the study design appropriate to address the stated objectives?

-Is the population clearly described and appropriate for the hypothesis being tested?

-Is the sample size sufficient to ensure adequate power to address the hypothesis being tested?

-Were correct statistical analysis used to support conclusions?

-Are there concerns about ethical or regulatory requirements being met?

Reviewer #1: yes

Reviewer #2: 1. Figures 4C, 6B, and 9B: Authors showed “log total pfu” along with Ct value of qRT-PCR. It is impossible to convert RNA copy number (a mixture of genomic RNA and mRNA) into plaque forming unit, which is the infectious virus unit. Please correct them into “Viral RNA copies” based on the comparison with standard curve using control viral RNA. The calculation should also be described in the materials and method.

2. Figures 5B, 8A, 8B should clarify the base of log.

3. Figure 9: The comparison of two values in the experiment with three or more groups should be made by ANOVA.

Reviewer #3: This paper describes the generation of a recombinant RVFV vaccine vector also designed to serve as a vaccine against BTV. The authors generated recombinant RVFV vaccine candidates encoding BTV4 VP2 or a truncated form of BTV4 NS1 called NS1Nt using reverse genetics. Both rescued viruses were shown to express their respective genes by immunofluorescence of infected cultures using specific antibodies. NS1Nt could be detected in infected cell lysates whereas degraded VP2 proteins were detected using immunoprecipitation. Growth kinetics and end titres as well as plaque morphologies were compared with the control. Stability of the viruses while passaging through cells was confirmed. 

Immunisation of Balb/C mice with the rescued viruses did not affect the antibody response against RVFV N. The VP2 immunisation showed a dose-dependent response on anti-N antibody production but a challenge with a lethal dose of RVFV showed survival of all mice given different doses and corresponding levels of viraemia with dose. Analysis of the VP2 antibody response in the same mice showed a dose-dependent VP2 antibody-specific response increasing over time and after a BTV4 boost. Neutralising antibodies also showed a dose-dependent response with a decrease after 21 days and a significant increase after the BTV4 boost. Only passive immunisation of IFNAR mice with serum from mice vaccinated with the VP2 vaccine and BTV4 virus boost resulted in their survival with correspondingly low viraemia levels. ICCS and ELISPOT results from serum of mice immunised with NS1Nt and a boost with BTV4 showed significant levels of cellular stimulation with lower levels from those receiving the NS1Nt only immunisation.

Sheep immunised with either the VP2 or NS1Nt vaccines showed significant levels of neutralising antibodies for VP2 and none for NS1Nt as well as high levels for RVFV neutralising antibodies for both vaccines. After a challenge with BTV8, lower levels of pyrexia, viraemia and anti-VP7 antibodies were observed for both VP2 and NS1Nt-vaccinated sheep.

The methods are extremely well written bar a few grammatical errors and it would be very easy to repeat these experiments - minor revision

**Results**

-Does the analysis presented match the analysis plan?

-Are the results clearly and completely presented?

-Are the figures (Tables, Images) of sufficient quality for clarity?

Reviewer #1: yes

Reviewer #2: 1. In sheep, the protective efficacy of two candidate vaccines against BTV was not clear, other than marginal reduction of viral replication judged from the RNA level. Apparently, this limited protective efficacy against BTV VP2 is due to the truncation of VP2 gene. Figure 2C showed that BTV VP2 proteins was largely truncated, although authors speculated that VP2 proteins were rapidly processed by specific proteases (line 602). The RVFV S-segment NSs gene cassette apparently has a size limitation to retain foreign gene. To demonstrate that majority of viral population retains the full-length of VP2 gene in the S-segment, Northern blot of chimeric S-segment RNA is essential. It is also important to characterize the sequence of remaining VP2 gene, to explain the 25kD size VP2.

2. AST and ALT values in Figure S3B may be included in the result, if partial efficacy of candidate vaccine is discussed.

Reviewer #3: The analysis mostly matches the plan. There are some results that are not discussed - see Summary and General Comments

Results are very clearly presented.

Images are not very good quality - it is hard to discern the differences between the plaque morphologies of figure 3b and fig 2a is also not very clear.

**Conclusions**

-Are the conclusions supported by the data presented?

-Are the limitations of analysis clearly described?

-Do the authors discuss how these data can be helpful to advance our understanding of the topic under study?

-Is public health relevance addressed?

Reviewer #1: No.

Major comments:

• The authors have used the incorrect term “chimeric” in the title and throughout the manuscript. The definition of chimera is a combination of two or more different species of which the parts are essential for the chimeric organism, here a chimeric virus. The described RVFV expressing GFP or a gene originating from BTV is not depending on NS1t or VP2 of BTV, and thus not a chimeric virus. Instead, the described viruses are viral vector vaccine candidates, in more detail duo of bivalent vaccine candidates potentially protecting animals against both RVF and BT. The term ”chimeric” should be removed throughout the manuscript and replaced by a an appropriate term, like bivalent vaccine or duo vaccine.

• The presented results do not justify the claim as stated in the title, since protection against BT has not been shown.

• “Conclusions/Significance” This reviewer has doubts about this prospect based on the presented data and the hurtles to be taken in order to achieve full protection for the added one, here BTV.

In addition:

1) Large parts of the world (Australia, Europe, Americas) are RVF-free at the moment. Here, authorities will be very reluctant to permit the use of a RVFV-vector based vaccine for protection against BT or any other disease. Easy serology-based monitoring to demonstrate freedom of RVF could be a problematic after use of such vector vaccine.

2) RVF endemic parts of the world will be reluctant too, since the start of an epidemic is harder to notice due to the same reason. Further, this RVFV vector is potentially transmittable to and by insect vectors. Another potential risk is exchange of the fragment with the foreign gene with wild type RVFV, although this will not lead to virulent RVFV. As a consequence, uncontrolled spread of RVFV-vectored vaccine or segments thereof cannot be excluded.

3) Complete protection against BT induced by the presented bivalent vaccine will require boost vaccination(s). The pre-existing immune responses against the viral vector, here attenuated RVFV or a high seroprevalence due to a previous RVFV infection, will reduce the outcome of boost vaccination against BTV.

4) In particular for Bluetongue, how to cope with multi-serotype situations in the field with such bivalent RVFV-vectored vaccine for BT?

Reviewer #2: Partial protection of sheep against BTV by rZH548-ΔNSs::VP2BTV-4, or rZH548- ΔNSs::NS1NtBTV-4 was not sufficiently supported by the experiment.

Reviewer #3: Conclusions are definitely supported by the data presented. There is some discussion of how the data can be helpful to advance understanding of the topic. Discussion touches on limitations of the analysis. Public health relevance is not addressed directly.

**Editorial and Data Presentation Modifications?**

Reviewer #1: Minor comments

• L59-60. Live-attenuated RVF vaccine strain Clone 13 is NOT completely safe, and thus not completely safe as viral vector for vaccine development!! Makoschey et al. Rift Valley Fever Vaccine Virus Clone 13 Is Able to Cross the Ovine Placental Barrier Associated with Foetal Infections, Malformations, and Stillbirths. PLoS Negl Trop Dis. 2016;10(3):e0004550.

• L97-98. Refences of these BTV-4 strains?? BTV-4M is a reassortant (L25-L523)!! Why using this strain as challenge inoculum?? How lethal/virulent is BTV-4M?

• L342-343. This is confusing upon Fig 2C. The arrows indicate very minor bands in lane 2 compared to other nonspecific?) bands and the 25 kDa band, while bands in the right lane are not linked to BTV proteins.

• L344-360 can be shortened and Fig 3 can be supp Figure as live-attenuated RVFVs are all similar, while the ancestor or empty live-attenuated RVFV is missing in this figure.

• L366-380. Indeed, CPE and IFI are more (most) important as these phenotypic assays show stability of gene expression. Alternatively, full genome sequencing should be performed to demonstrate overall genetic stability.

• L380. I assume the rescued virus is passed on cell culture and not the (infected/transfected) cells.

• L383. RVFV or RVF virus!!

• Sup Fig 2 and Fig 4: D.O.: Likely optical density (OD).

• L389-399. The characteristics of the ancestor empty live-attenuated rescued RVFV are likely known, and should be used as control to compare with candidate to study the effect of the VP2 insert.

• L410-428. Unfortunately, the (n)Abs against VP2 seems to be temporary. Lasting protection, if any, is most unlikely after one shot vaccination.

• L436-464. (Fig 6.) This experiment is very artificial, however, indicative that something happened. Still, not very important and should be included in supp. Data.

• L508-510. Indeed, the GFP variant intended to induce a lower humoral response as can be seen in both Figure 8C and for both tested days in Figure 8B.

• L525-526. The lower mean temperature for the GFP variant is obvious from day 3 to day 6. I don’t know why. Still, the onset of pyrexia is very similar for all three groups.

• L529-533. This reviewer did not understand this. All three groups reached Ct values corresponding to >3 log pfu??

• L536-538. This reviewer agrees with the statement that all data together suggest that vaccinated groups showed a very slight difference compared to the GFP control group. 

• Supp fig 3A: indeed a very very small difference if any.

• Supp fig 3B: differences are very marginal, if any could be observed!

• In addition, BTV4-M did not cause mortality or severe clinical signs as shown for the (control) sheep vaccinated with the GFP variant.

• L557-581. Taken together, it is questionable whether RVFV is a good choice as viral vector vor vector vaccines for other ruminant diseases, see above.

• L582-620. I agree that these BTV antigens are the most promising. However, previous studies showed a variety of results suggesting that each antigen will not induce complete protection. At least this has been confirmed in this study. Even more, the results are very disappointing. Likely this is caused by truncated (NS1) and degraded (VP2) proteins. Incomplete proteins mostly show a high turnover, and thus are not presented well to the immune system resulting in a poor response as here shown.

• In general, the discussion section should be shortened.

Reviewer #2: "data not shown" is not acceptable in this journal.

Reviewer #3: Minor revision

**Summary and General Comments**

Reviewer #1: This manuscript describes the rescue of live-attenuated RVFV by reverse genetics containing and expressing of truncated NS1- or VP2 gene of BTV. In addition to in vitro analyses to confirm expression of the foreign genes, immunization trials in mice and sheep were performed and extensively monitored. In addition, vaccinated ruminant target hosts of RVFV and BTV (sheep) were challenged by BTV4 infection. The authors hypothesize protection of target animals against both diseases by vaccination with such bivalent vaccine.

In general, the results are well presented, but the conclusions are more than once “wishful thinking”. In summary, the results are interesting, although disappointing with respect to vaccine development for Bluetongue. Overall, the manuscript must be significantly rewritten with major revisions, and in particular the interpretations of the results must be strongly weakened before it could be reconsidered for publication in PLOS Neglected Tropical Diseases.

Reviewer #2: The manuscript entitled “Chimeric Rift Valley fever viruses encoding bluetongue virus (BTV) antigens can provide immunity and partial protection against a BTV-4 challenge in sheep” by Moreno S et al. describes that the vaccination with the recombinant ZH548 strain (rZH548) of Rift Valley fever virus (RVFV), which encodes Bluetongue virus (BTV) outer capsid protein VP2 in place of RVFV NSs gene in the S-segment (“rZH548-ΔNSs::VP2BTV-4), confers protective immunity in mice challenged with pathogenic RVFV 56/74 strain. Although immunocompetent mice are refractory to BTV infection, passive transfer of sera derived from BALB/c mice vaccinated with rZH548-ΔNSs::VP2BTV-4 and received an additional booster with 1x10^5 PFU of BTV-4 conferred protection of IFNAR-/- A129 mice against BTV serotype 4 (BTV-4). This study also tested the cellular immune responses to another recombinant RVFV (“rZH548- ΔNSs::NS1NtBTV-4), which encodes BTV nonstructural protein NS1 (partial expression of the N-terminus) in place of RVFV NSs gene in the S-segment, via intracellular cytokine staining or ELISPOT assay. Again, the booster with BTV-4 showed a higher level of CD8 T cell stimulation. The protective efficacy of 1x10^7 pfu doses of rZH548-ΔNSs::VP2BTV-4, or rZH548- ΔNSs::NS1NtBTV-4 against the virulent BTV-4 strain was further tested in sheep at 21 days post vaccination. None of those candidate vaccines showed protection against BTV-4 challenge (pyrexia, viremia), whereas authors claimed partial protection of sheep, due to slightly increased Ct in the qPT-PCR for BTV (~ 1 log10 PFU equivalent). The expression of BTV VP7 derived from challenge virus was apparently lower in vaccinated sheep than the control. Supporting Figure 3B indicates that the vaccination with rZH548- ΔNSs::NS1NtBTV-4 apparently prevented the increase of ALT upon BTV-4 challenge. Overall, the study extensively characterized the immunogenicity and efficacy of two candidate chimeric vaccines for RVF and Bluetongue, which will provide useful information for others considering a similar approach using RVFV NSs gene cassette for heterologous antigen expression. Potential weakness of this study is a lack of promising protective efficacy against Bluetongue.

Reviewer #3: This manuscript covers describes a large body of work and the idea of making a dual vaccine using the reverse genetics RVFV construct is novel. There are a number of places where some bits of information are lacking. See specific comments below:

Introduction

The introduction is well written and just needs correction of a few grammatical errors – these are indicated as comments in the attached pdf.

Methods

Under the section entitled ‘plasmid construction’ – it would be a good idea to refer to 1A and 1B in the appropriate places in the text so that the reader knows which part of the figure to look at.

End of line 126 – ‘y’ - typographical error?

Results

Figure 1 legend – Restriction enzyme site names – these are italicised I the figure legend but not in other parts of the paper – choose one format and keep it consistent

Line 330 – insert the size in kDa of the NS1Nt band

Lin 331 – it wasn’t detected at all! Ie was not detected

Lines 330-332 – you mention in the methods that you did immunoprecipitation for MP12 and NS1Nt as well - what did the MP12 and NS1Nt look like in the immunoprecipitation analysis?

What are the other bands in the BTV4 lane - where is VP7 and NS1?

Line 340 – you should probably indicate that you used anti-VP2 as well even though this didn’t work.

Fig3B – the figure quality is not great – I had difficulty in seeing the plaque phenotype difference between GFP and VP2/NS1Nt.

Lines 374-375 - I think these are the wrong way around in the text - according to the figure NS1 is normalised to the L segment and VP2 to the S segment.

FigS2 legend – you should indicate what the mock inoculum was. Also, what does D O mean? Maybe you mean OD?

Line 390 – 392 – I was wondering, what about the NS1Nt? Does it have the same effect as VP2 on the N antibody response? 

Line 498 – it would have been interesting to include a group having a co-inoculation both vaccine candidates.

Line 520 – Fig 8A legend - I think the orange circles are meant to be orange squares according to the figure legend? 

Fig8A – what does the first black arrow on day 4 indicate?

Line 503-504 - How do you explain the increase of all of the seroneutralisation titres including the GFP control after the BTV4 challenge?

You only looked at the response to BTV VP2 and RVFV – what about NSNt ie doing ICCS and/or ELISPOT? Why was it not possible to do this?

Supp Fig 3B – if you present all of the biochemical parameters in the figure, you should mention all of them in the results section ie what about tp, ggt and alb - you don't mention these. 

Discussion

Line 589 – it would have been interesting to look at NS1 antibodies in the sheep challenge trial experiment too.

Line 682-683 – While I know that it is important to determine what each vaccine candidate contributes to immunity and protection why didn’t you have a group of sheep which received a combination of the 2 recombinant vaccines?

Fig S2 figure legend – you should mention what the mock immunogen is in the figure legend.

PLOS authors have the option to publish the peer review history of their article (what does this mean?). If published, this will include your full peer review and any attached files.

Reviewer #1: No

Reviewer #2: No

Reviewer #3: Yes: Ann Meyers
---

## [Decision Letter · Decision Letter 1]

20 Oct 2020

Dear Dr Brun,

Thank you very much for submitting your manuscript "Recombinant Rift Valley fever viruses encoding bluetongue virus (BTV) antigens: immunity and efficacy studies upon a BTV-4 challenge." for consideration at PLOS Neglected Tropical Diseases. As with all papers reviewed by the journal, your manuscript was reviewed by members of the editorial board and by several independent reviewers. The reviewers appreciated the attention to an important topic. Based on the reviews, we are likely to accept this manuscript for publication, providing that you modify the manuscript according to the review recommendations. 

The reviewers from your original submission have reviewed your manuscript and identified some areas that require clarification. Most of these comments are relatively minor, bit do need to be addressed.

Sincerely,

Michael R Holbrook, PhD

Associate Editor

Amy Gilbert

Deputy Editor

The reviewers from your original submission have reviewed your manuscript and identified some areas that require clarification. Most of these comments are relatively minor, bit do need to be addressed.

Reviewer's Responses to Questions

**Key Review Criteria Required for Acceptance?**

**Methods**

-Are the objectives of the study clearly articulated with a clear testable hypothesis stated?

-Is the study design appropriate to address the stated objectives?

-Is the population clearly described and appropriate for the hypothesis being tested?

-Is the sample size sufficient to ensure adequate power to address the hypothesis being tested?

-Were correct statistical analysis used to support conclusions?

-Are there concerns about ethical or regulatory requirements being met?

Reviewer #1: yes

Reviewer #2: Authors addressed specific points appropriately. This reviewer has no further comments.

Reviewer #3: yes

**Results**

-Does the analysis presented match the analysis plan?

-Are the results clearly and completely presented?

-Are the figures (Tables, Images) of sufficient quality for clarity?

Reviewer #1: yes

Reviewer #2: Authors addressed specific points appropriately. This reviewer has no further comments.

Reviewer #3: I still think figure 2c needs clarification - maybe look at the figure legend and the results presented in the text.

**Conclusions**

-Are the conclusions supported by the data presented?

-Are the limitations of analysis clearly described?

-Do the authors discuss how these data can be helpful to advance our understanding of the topic under study?

-Is public health relevance addressed?

Reviewer #1: yes

Reviewer #2: Authors addressed specific points appropriately. This reviewer has no further comments.

Reviewer #3: I agree with one of the other reviewer's comments about the limitation of introducing such a vaccine into RVFV-free regions or RVFV endemic regions. The authors should probably include a sentence or two on these limitations.

**Editorial and Data Presentation Modifications?**

Reviewer #1: Accept

Reviewer #2: Authors addressed specific points appropriately. This reviewer has no further comments.

Reviewer #3: Lines 130-132 – it’s still not quite clear what you did here – please clarify

Lines 313-321 – you need to refer to figure 1a and b separately

Line 320 – which 5 plasmids?

Lines 347-349 – it’s still hard to distinguish between cytoplasm and aggregated bodies – arrows in the figure might help

Line 354 – supp fig 1A or 1B?

Line 351 – isn’t the size of BTV VP2 above 80 kDa? How can you say it was barely detectable if you don’t show this the appropriate MW range in fig 2b?

Lines 367-368 – in the methods, you indicate the anti-serum used was mouse anti-BTV-4 NS1/VP2/VP7 – how can this detect VP5?

Lines 368- 370 – I find this confusing?

Lines 405-406 – in this section – any reason why you didn’t use the rZH548-ΔNSs as a control?

**Summary and General Comments**

Reviewer #1: Review report manuscript no. PNTD-D-20-00574R1

Title: “Recombinant Rift Valley fever viruses encoding bluetongue virus (BTV) antigens: immunity and efficacy studies upon a BTV-4 challenge”.

Authors: Sandra Moreno, Eva Calvo-Pinilla, Stephanie Devignot, Friedemann Weber, Javier Ortego, and Alejandro Brun.

This revised manuscript describing the rescue of live-attenuated RVFV by reverse genetics expressing truncated NS1- or VP2 gene of BTV has been significantly improved.

The authors are speculating about the potential of this RVF vectored vaccine platform for more endemic ruminant diseases and therewith protecting against RVF. This is an attractive approach to appeal vaccination for this zoonotic disease.

In summary, the results are interesting as the vectored vaccine platform is safe but the efficacy requires significant improvement. 

The manuscript is now acceptable for publication in PLOS Neglected Tropical Diseases.

Reviewer #2: Revised manuscript clarified the expression of full-length of VP2 mRNA by RT-PCR. Although it is still possible that proposed recombinant RVFV expressing BTV VP2 is a mixture of truncated S-segment and full-length S-segment, authors described the possibility of VP2 truncation in the text, regardless of true mechanism. In terms of protective efficacy, the candidate vaccine does not confer good protection against BTV challenge, which might indicate the insufficient expression of BTV antigens from RVFV NSs cassette. Nevertheless, the study overall carefully described full set of experiment for potential candidate vaccine, which will be sufficient content for the publication at this point.

Reviewer #3: (No Response)

PLOS authors have the option to publish the peer review history of their article (what does this mean?). If published, this will include your full peer review and any attached files.

Reviewer #1: No

Reviewer #2: No

Reviewer #3: No
---

## [Editor Report · Decision Letter 2]

2 Nov 2020

Dear Dr Brun,

We are pleased to inform you that your manuscript 'Recombinant Rift Valley fever viruses encoding bluetongue virus (BTV) antigens: immunity and efficacy studies upon a BTV-4 challenge.' has been provisionally accepted for publication in PLOS Neglected Tropical Diseases.

Please carefully review your manuscript prior to/during the proofing stage. There are a handful of typographical errors that should be addressed. The PLoS Editorial team does not correct grammar/spelling errors.

Best regards,

Michael R Holbrook, PhD

Associate Editor

Amy Gilbert

Deputy Editor

---

## [Editor Report · Acceptance letter]

25 Nov 2020

Dear Dr Brun,

We are delighted to inform you that your manuscript, "Recombinant Rift Valley fever viruses encoding bluetongue virus (BTV) antigens: immunity and efficacy studies upon a BTV-4 challenge.," has been formally accepted for publication in PLOS Neglected Tropical Diseases.

Best regards,

Shaden Kamhawi

co-Editor-in-Chief

Paul Brindley

co-Editor-in-Chief
